# HOBA: Higher-Order Block-Diagonal Attention Unrolling for Transformer

## Abstract

Transformers with 2D self-attention are powerful but computationally intensive, specifically for long sequences due to their quadratic complexity. Therefore, sparse attention methods attempt to alleviate this cost by limiting attention patterns. However, they often compromise explainability and fail to generalize well to global dependencies. Therefore, we propose **H**igher-**O**rder **B**lock-Diagonal **A**ttention (**HOBA**), a novel transformer variant that models triplet interactions utilizing 3D attention tensors and block-diagonal unrolling. HOBA can capture richer patterns within and across blocks while efficiently modeling long-range dependencies without high computational cost. We use knowledge distillation with RoBERTa as the teacher to train the HOBA student model. We evaluate HOBA on five NLP tasks across eight benchmark datasets, comparing it against Full-3D (without block or cross-block), standard 2D attention, and sparse mechanisms including Longformer, BigBird, Local, and Dilated attention. We further isolate the contributions of block structure and higher-order interactions, confirming HOBA's superiority over both dense and sparse baselines. We also demonstrate that allowing cross-block interaction yields significant accuracy gains by enhancing long-range token dependencies.

## 1 Introduction

Computational linguistics has made significant progress using transformer architectures (Liu et al., 2023; Vaswani et al., 2017). However, these transformer architectures encounter a crucial scalability issue, especially when dealing with long sequence analysis. The root of the problem lies in the conventional self-attention mechanism, which suffers from quadratic complexity. This makes it inefficient to process longer sequences or real-time applications such as document processing and dialogue systems(Brown et al., 2024; Bai et al., 2024). This inefficiency is because 2D attention computes dense interactions between all pairs of tokens in the sequence. Recent sparse attention techniques such as Sparse Transformer (Child et al., 2019), Longformer (Beltagy et al., 2020), BigBird (Zaheer et al., 2020), Reformer (Kitaev et al., 2020), and Linformer (Wang et al., 2020) lessen this by eliminating token pairs selectively while aiming to preserve key contextual dependencies. These sparse transformers achieve efficiency by limiting attention computations to local windows, low-rank projections, or global tokens, thereby approximating dense interactions while preserving important contextual information. However, these methods focus primarily on computational efficiency while maintaining the fundamental pairwise interaction paradigm.

The transparency of these models is limited by their ability to produce approximated or fragmented attention maps that are difficult to analyze. Additionally, many of these architectures are developed with specific applications in mind, such as Longformer for question-answering or summarization (Sun et al., 2025) and Linformer for classification (Wang et al., 2020), resulting in limited generalizability. Besides this, block-based architectures such as Reformer (Kitaev et al., 2020) also struggle to model long-range dependencies due to their localized design. Despite comprehensive attention efficiency research, only few attempts to pay higher-order(HO) attention due to its high cubic computational cost and difficulty in stable training. Models like Jump Attention (Zhou et al., 2022) and Kronecker-Attention(Gao et al., 2020) explore this direction; however, they remain largely theoretical or tailored to niche tasks. So the question comes: ***Is it possible to design an attention mechanism that can better capture HO dependencies while still being computationally efficient and expressive?***

In this work, we propose a paradigm shift from traditional pairwise attention to HO attention that models contextual dependencies among token triplets. We introduce **H**igher-**O**rder **B**lock-Diagonal **A**ttention (**HOBA**) for sequence-wise overlapping block decomposition using a diagonal structure to capture triplet-level interactions efficiently. However, naive 3D attention is infeasible due to its cubic complexity. Thus, our approach reshapes attention through local HO interactions in the conditioned sparsity blocks. Basically, HOBA computes a 3D attention tensor in a localized way by focusing on token triplets within block-diagonal regions. This reduces the complexity from cubic to near-linear while maintaining expressiveness. HOBA assigns higher attention to semantically coherent token triplets by demonstrating its power to capture richer contextual dependencies beyond standard pairwise 2D interactions. Additionally, HOBA's cross-block mechanisms allow global context modeling without sacrificing the efficiency of local processing. We evaluate HOBA on five representative language understanding tasks: sentiment analysis, news classification, question classification, natural language inference, and question answering. We measure HOBA's practical efficiency by evaluating its FLOPs, token processing speed, memory utilization, and accuracy. HOBA is modular and highly extensible, and it can be easily incorporated into other transformer-based models. Its tunable block size and overlap make it versatile for various tasks and resources. **The contributions of this work are as follows:**

1. We design HOBA, *the first scalable triplet-based attention mechanism that models HO interactions utilizing localized block-diagonal structures*. It dramatically reduces the cubic cost of 3D attention while maintaining attention granularity by enabling deployment in real-world tasks. We use knowledge distillation from a pre-trained RoBERTa teacher to train a smaller student(HOBA) that learns efficiently through soft labels and attention alignment.

2. We incorporate a cross-block mechanism into HOBA that forms token triplets across disjoint blocks. *It allows HOBA to capture global dependencies without sacrificing efficiency*. We show that utilizing cross-block interaction can improve accuracy by up to 13% and reduce loss by up to 0.22, which proves its essential role in capturing long-range dependencies.

3. We evaluate HOBA across eight datasets on five different NLP tasks and compare with recent sparse attention models and standard 2D RoBERTa. HOBA consistently performs better with gains of up to +3.5% by offering the fastest training time up to $2.6\times$ faster compared to strong baselines. Also, it remains robust across different layer depths and sequence lengths by demonstrating both effectiveness and generalization in diverse NLP settings.

**Why Triplets Over Pairs?** HOBA differs from standard 2D attention not because it uses a 3D tensor, but because the 3D tensor explicitly models third-order token interactions that cannot be captured by pairwise dot-product attention. In conventional Transformers, each attention score depends only on the relation between two tokens, limiting the model to binary dependencies (e.g., subject→verb or adjective→noun). In contrast, HOBA's triplet attention tensor $A_{ijk} = Q_i^\top (K_j \circ K_k)$ incorporates the joint context formed by a pair of keys $(j, k)$, enabling the model to compute attention scores based on interactions of two context tokens simultaneously. This provides access to richer relational structures that naturally arise in language, such as scope resolution and negation ("not really good"), multi-modifier composition ("very highly unlikely"), and compound noun cohesion ("interest rate policy"), all of which depend on three-way configurations rather than isolated token pairs. Whereas 2D attention must approximate these higher-order effects across multiple layers, HOBA can express them within a single layer, offering a stronger inductive bias for structured multi-token reasoning.

## 2 RELATED WORKS

**Higher-Order and Sparse Attention.** The quadratic cost of standard self-attention has motivated extensive work on structured approximations. Sparse attention models such as Longformer and BigBird introduce local windows and global tokens to lessen computational complexity while maintaining contextual awareness (Beltagy et al., 2020; Zaheer et al., 2020). Similarly, Linformer offers low-rank projections to facilitate attention computation (Wang et al., 2020), while Performer and Routing Transformer replace traditional dot products with kernel-based and routing-based approximations for improved scalability (Choromanski et al., 2020; Roy et al., 2021). Jump Self-Attention (JAT) extends this direction by capturing HO dependencies via spectral convolution, enhancing performance in tasks like natural language understanding (Zhou et al., 2022). Likewise, WIGRAPH

presents a trainable layer that learns global word-level interactions, improving both interpretability and predictive accuracy in NLP models (Sekhon et al., 2023). Another work proposes Sparse Self-Attention Fine-tuning (SSAF) that can replace softmax with a controllable sparse transformation to improve the performance and interpretability of BERT fine-tuning (Cui et al., 2019). Beyond sparsity, recent HO transformers reduce computational overhead using Kronecker product factorization and kernelized attention approximations (Omranpour et al., 2024). Similarly, Graph-Induced Attention goes further by embedding syntactic or semantic graph structures directly into the attention mechanism by enabling stronger inductive bias and structural awareness (Hong et al., 2022). A conceptual comparison of these higher-order attention methods is provided in **Appendix A.7**.

**Standard Attention and Knowledge Distillation.** Transformers like BERT, RoBERTa, and T5 have become foundational in core NLP tasks like sentiment analysis (Nuci et al., 2024), natural language inference (Raparthy et al., 2023), and topic classification (Kosar et al., 2023). Each task requires different modeling strengths; sentiment analysis needs polarity sensitivity (Nuci et al., 2024), NLI demands contextual reasoning (Raparthy et al., 2023), and news classification favors topical abstraction. To address this diversity, researchers have proposed task-aware attention mechanisms (Benarab & Gui, 2022; Roy et al., 2021), modular tuning methods like adapters (Pfeiffer et al., 2020), and data-centric augmentation techniques (Chai et al., 2024). However, challenges persist in domain generalization, compositional reasoning, and inference efficiency (Patil & Jadon, 2025). Knowledge distillation processes have been widely adopted to mitigate these challenges. These methods compress large transformer models by training compact students on softened outputs from teacher models using temperature scaling and Kullback-Leibler(KL) divergence (Lu et al., 2022). In practice, teacher-student setups align final logits or internal attention patterns through KL or MSE losses, often blending hard and soft supervision (Aguilar et al., 2020; Chen et al., 2021). Recent methods use progressive stages to gradually shrink the model (Yao et al., 2024), bias-aware filtering (Zhang et al., 2025), or cooperative schemes (Livanos et al., 2024). Specialized distillation has also been explored in sentiment and emotion-aware models (Li et al., 2025; Ma et al., 2023).

## 3 BLOCK-DIAGONAL UNROLLING FRAMEWORK

### 3.1 PROBLEM FORMULATION

Let $X = [x_1, x_2, \ldots, x_L] \in \mathbb{R}^{L \times d}$ be an input sequence of $L$ tokens, where each token embedding $x_i \in \mathbb{R}^d$. The goal is to learn a transformation $f_\theta(X)$ that produces a contextual representation $H \in \mathbb{R}^{L \times d'}$ for a variety of natural language tasks. In transformer models, $f_\theta$ consists of stacked layers of feedforward networks and self-attention. The self-attention module allows each token to collect contextual information from the entire sequence.

Given projections $Q = XW_Q$, $K = XW_K$, and $V = XW_V$ with learnable parameters $W_Q, W_K, W_V \in \mathbb{R}^{d \times d_k}$, the attention output is computed as

$$H = \text{Attention}(Q, K, V) = \text{softmax}\left(\frac{QK^\top}{\sqrt{d_k}}\right) V \qquad (1)$$

The attention matrix $A \in \mathbb{R}^{L \times L}$ that results from this traditional method captures all paired token interactions. However, it poses two major challenges. First, the attention operation has quadratic complexity $\mathcal{O}(L^2 d)$ in both time and memory, which restricts scalability. Second, the fully dense structure lacks inductive bias, which makes it challenging to interpret and potentially redundant. To resolve these issues, we propose a reformulated attention operator $\widetilde{H} = \mathcal{F}_{3D}(X)$, where $\mathcal{F}_{3D}$ captures higher-order(3D) interactions utilizing structured sparsity. Our 3D attention mechanism replaces the dense 2D attention map with a block-diagonal structure that improves both efficiency and performance.

### 3.2 PROPOSED 3D BLOCK-DIAGONAL ATTENTION: HOBA

Our 3D attention mechanism, HOBA, generalizes standard 2D self-attention by introducing a third-order interaction space. We model interactions among token triplets $(x_i, x_j, x_k)$ instead of computing attention over all token pairs $(x_i, x_j)$ and capture richer relational patterns that go beyond simple

pairwise dependencies. The 3D block-diagonal attention mechanism from end to end is detailed in Figure 1.

Given an input sequence $X = [x_1, x_2, \ldots, x_L] \in \mathbb{R}^{L \times d}$, 3D attention creates a tensor $\mathcal{A} \in \mathbb{R}^{L \times L \times L}$ such that each component $\mathcal{A}_{i,j,k}$ encodes the interaction score among tokens $x_i, x_j, x_k$. For simplicity, the head dimension is omitted, and below Eqn. (2) is applied independently per head. We compute the contextual output ($h_i$) for each token by aggregating over this triadic space:

$$h_i = \sum_{j=1}^{L} \sum_{k=1}^{L} \alpha_{i,j,k} \cdot g(x_j, x_k) \tag{2}$$

where $g(x_j, x_k)$ is a function that combines the two context tokens, and $\alpha_{i,j,k}$ is the normalized attention score over $j$ and $k$. This approach allows richer semantic modeling, such as resolving phrase-level sentiment or subject-modifier-object interactions. We approximate $h_i$ via block-level contextual outputs $C^{(m)}$, which are later aggregated into the final sequence representation $O$. However, 3D attention faces practical limitations despite its modeling power. It is inherently more expensive in memory and computation than standard 2D attention, and computing full 3D tensors naively is intractable for long sequences.

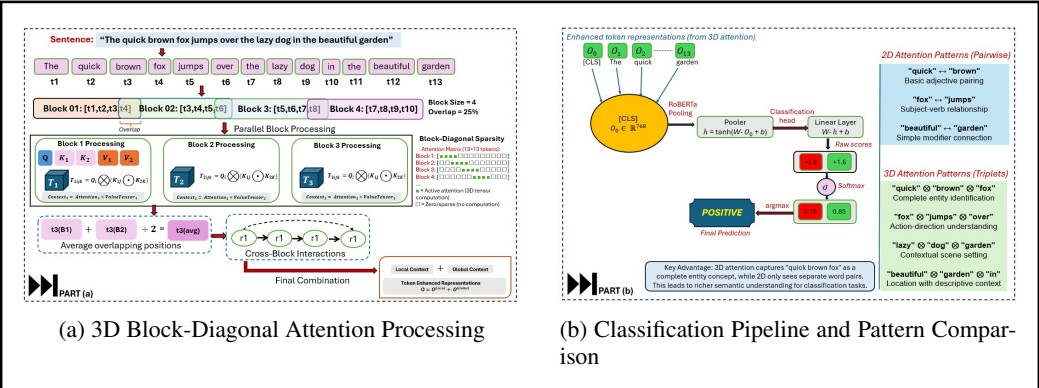

(a) 3D Block-Diagonal Attention Processing    (b) Classification Pipeline and Pattern Comparison

Figure 1: Complete HOBA unrolling Framework. (a) Once the input sentence is tokenized, we split the tokens into several blocks and process the 3D tensor parallelly within a block-diagonal sparsity pattern that eliminates redundant full-matrix computations. The block-diagonal attention matrix shows how our method gains computational efficiency by restricting attention to local block regions while preserving sparse zero-computation areas outside the diagonal blocks. Later, we do overlap averaging and compute cross-block interactions. Finally, we get the enhanced representation of the tokens with lowered complexity from O(n²) standard attention to O(nb²) block-diagonal attention. (b) Enhanced token representations flow through RoBERTa pooling to the final prediction. Additionally, we show how 3D attention patterns capture richer semantic relationships than standard 2D attention, leading to improved classification performance.

Due to the computational complexity of simultaneously computing attention scores across query vectors and multiple key vector pairs in our HO attention mechanism, we use PyTorch's *einsum* operation to implement HO interactions for contracting queries with pairs of keys to form multidimensional attention tensors that capture triplet relationships. It enables the compact implementation of complex tensor operations that would otherwise require higher multiple matrix multiplications and reshaping operations.

### 3.2.1 BLOCK-DIAGONAL UNROLLING

HOBA uses HO interactions in a structured and localized manner to overcome the cubic computational complexity of 3D attention. Therefore, instead of operating on the full $L \times L \times L$ attention cube, which grows cubically with input length, we split the sequence $X \in \mathbb{R}^{L \times d}$ into $B$ non-overlapping blocks of size $b$, such that $L = B \cdot b$.

Suppose that our sequence has $L = 12$ tokens and we set block size $b = 4$, then we can express it as:

$$X^{(1)} = [x_1, x_2, x_3, x_4], \qquad X^{(2)} = [x_5, x_6, x_7, x_8], \qquad X^{(3)} = [x_9, x_{10}, x_{11}, x_{12}].$$

we compute a local 3D attention tensor $\mathcal{A}^{(b)} \in \mathbb{R}^{b \times b \times b}$ for each block $X^{(i)}$. Each entry $\mathcal{A}^{(b)}_{j,k,\ell}$ models the interaction between triplets $(x_j, x_k, x_\ell)$ within the block. We traverse diagonal slices of the cube $\mathcal{A}^{(b)}$ rather than iterating over all key pairs $(j, k)$, where $j + k = \text{constant}$. This improves computational efficiency through parallelism and better cache locality. The block-diagonal structure reduces the computational complexity from $\mathcal{O}(L^3)$ to $\mathcal{O}(B \cdot b^3) = \mathcal{O}(Lb^2)$, where $b \ll L$ (details on accounting for overlap and Cross-Block costs are discussed in **Appendix A.6**). This formulation achieves linear scaling with respect to sequence length for a fixed block size $b$, while preserving the expressive capacity of HO interactions within local regions (more details in **Appendix A.3**).

### 3.2.2 OVERLAPPING BLOCK STRUCTURE

We expand our block partitioning strategy by introducing overlapping blocks to allow smooth information flow across block boundaries. Each block $X^{(m)}$ now spans a wider window specified by an overlap ratio $\lambda \in (0, 1)$. Specifically, block $m$ covers the token positions:

$$[m \cdot b(1 - \lambda), \; m \cdot b(1 - \lambda) + b) \tag{3}$$

This overlapping scheme permits contextual blending across adjacent blocks without substantially raising the memory footprint. For any token $x_i$ that arises in multiple overlapping blocks, we average its attention outputs from each block to get the final representation:

$$o_i = \frac{1}{|B_i|} \sum_{m \in B_i} o_i^{(m)} \tag{4}$$

where $B_i$ represents the set of blocks that include token $x_i$, the overlap-aggregated token output is $o_i$, and $o_i^{(m)}$ is the block-specific output.

### 3.2.3 CROSS-BLOCK INTERACTION MECHANISM

Although local 3D attention captures rich intra-block dependencies, it cannot model global structure. Thus, we introduce a cross-block attention mechanism based on block-level summaries to address this. For each block $X^{(m)}$, we calculate a representative vector $r^{(m)} \in \mathbb{R}^d$ by averaging all token embeddings within the block:

$$r^{(m)} = \frac{1}{b} \sum_{i=mb}^{(m+1)b-1} h_i \tag{5}$$

We then calculate cross-block attention scores between all block representatives utilizing scaled dot-product attention:

$$\mathcal{A}^{(\text{cross})}_{m,m'} = \text{softmax}\left( \frac{(r^{(m)})^\top r^{(m')}}{\sqrt{d}} \right) \tag{6}$$

Finally, the output representation for token $x_i$ is updated by fusing local 3D attention with global block-level context:

$$o_i = o_i^{(\text{local})} + \sum_{m'=0}^{B-1} \mathcal{A}^{(\text{cross})}_{m,m'} \cdot r^{(m')} \tag{7}$$

where $m = \lfloor i/b \rfloor$ defines the block index corresponding to token position $i$. Here $o_i$ denotes the per-token output, and the global matrix $O = [o_1, \ldots, o_L]$ is returned as the final HO representation.

### 3.3 KNOWLEDGE DISTILLATION WITH HIGHER-ORDER STUDENT

We use knowledge distillation to facilitate stable training of HOBA while preserving architectural compatibility with RoBERTa. Precisely, we use a 12-layer RoBERTa-base model, fine-tuned on the target datasets with standard 2D self-attention, as the teacher. The teacher provides soft targets that capture class probabilities and inter-class similarities. It is often more informative than hard one-hot labels, specifically when training smaller student models. Our HOBA student model replaces the

teacher's 2D attention with block-diagonal HO interactions but inherits embedding and projection weights to ensure initialization consistency. While it is true that distilling from a 2D teacher constrains the absolute upper bound of performance, this setup provides a fair and controlled comparison by retaining the supervision signal identical across 2D and HOBA students. Moreover, distillation stabilizes training by preventing overfitting when moving from conventional quadratic attention to HO tensorized attention. The total loss integrates cross-entropy with ground-truth labels and KL divergence between the softened student and teacher distributions can be defined as

$$\mathcal{L}_{\text{total}} = (1 - \alpha) \cdot \mathcal{L}_{\text{CE}}(z_S, y) \ + \ \alpha \cdot \tau^2 \cdot \mathcal{L}_{\text{KL}}\big(\text{softmax}(z_S/\tau) \,\|\, \text{softmax}(z_T/\tau)\big), \qquad (8)$$

where $\mathcal{L}_{\text{CE}}$ is the cross-entropy loss with ground-truth labels $y$, $\mathcal{L}_{\text{KL}}$ is the KL divergence between teacher and student distributions, $\tau$ is the temperature for smoothing, and $\alpha \in [0, 1]$ is the balancing coefficient.

**Datasets**   We evaluate our HOBA method across diverse NLP tasks to demonstrate its general applicability(See Details in **Appendix A.2**, Table 8). We focus on five primary task categories: sentiment analysis(SA), news classiciation(NC), qestion classification(QC), natural language inference(NLI), and question answering(QA). These datasets express different levels of complexity, from binary sentiment classification to multi-class reasoning tasks, by facilitating a thorough evaluation of our method's effectiveness across diverse linguistic phenomena.

## 4   RESULTS ANALYSIS

### 4.1   EXPERIMENTAL SETUP

We compare vanilla RoBERTa models with our proposed HOBA variants under identical training configurations to ensure fairness. Both models use 3-layer and 6-layer transformer backbones with hidden size 768, 12 attention heads, and a feed-forward dimension of 3072. Specifically, we use either a 3-layer or 6-layer architecture with a small batch size of 8 and a maximum input sequence length of 64 tokens (for 3-layer models) or 128 tokens (for 6-layer models). In HOBA, standard 2D multi-head self-attention is replaced by block-diagonal higher-order attention, with block sizes $b \in \{16, 64\}$ and overlap ratio 25%. Embeddings and classification heads are initialized from a pretrained RoBERTa-base teacher to guarantee architectural compatibility. The teacher is a 12-layer RoBERTa-base model fine-tuned on the target task and kept frozen during student training. Knowledge distillation is applied using the combined cross-entropy and KL divergence losses with balancing factor $\alpha = 0.3$ and temperature $\tau = 2.0$. All models are trained with AdamW (learning rate $1 \times 10^{-5}$, weight decay 0.01), a linear decay schedule with 10% warmup, dropout rate 0.1, batch size 8, and early stopping after 3 epochs without validation improvement. We train for 5 epochs depending on the task, and average results across three random seeds (seed = 42) to report mean and standard deviation. Finally, we report student model performance to highlight the effectiveness of HOBA compared to the vanilla baseline(see Table 2). We use green color to highlight the best performer in tables.

### 4.2   FROM INFEASIBLE FULL-3D ATTENTION TO SPARSE AND FEASIBLE HOBA

Before scaling to large benchmarks, we first conduct a sanity check in a small-scale setting with a maximum sequence length of 64. This setup helps us isolate the source of HOBA's performance gains by evaluating the feasibility of HO attention in its pure form. In particular, we evaluate full 3D attention without block sparsity or cross-block interactions on two lightweight datasets: SST-2 (10k samples) and TREC (5k samples). This setting is intentionally simple: the sequence length is short, the number of classes is small, and the datasets are relatively limited in size. The results in Table 1 show that full 3D attention underperforms, reaching only 82–85% accuracy, and significantly below both vanilla 2D and HOBA across both 3-layer and 6-layer models. In contrast, HOBA consistently recovers accuracy, reaching 86–94% and closely matching or even surpassing the 2D baseline.

Later, in large benchmarks, HOBA achieves superior or comparable accuracy (see Table 2) in 8 out of 10 comparisons. For example, HOBA reaches 97% accuracy on SQuAD with 3 layers and 98% with 6 layers, consistently outperforming the 2D baseline. On MNLI, HOBA improves from 62% (2D) to 64% (3-layer) and further to 77% (6-layer), showing benefits on more complex reasoning

Table 1: SST-2 and TREC results with *Full-3D*, *HOBA*, and *2D*. Results are mean±std over three runs. Full-3D underperforms even at $L=64$, while HOBA recovers accuracy with better stability and efficiency. HOBA outperforms Full-3D but remains comparable to 2D(due to limited sample size).

| Dataset | Classes | Accuracy | | | Total Loss | | |
|---------|---------|----------|------|-----|------------|------|-----|
| | | Full-3D | HOBA | 2D | Full-3D | HOBA | 2D |
| **3-Layer Model** | | | | | | | |
| SST-2 | 2 | $83.4_{\pm 0.8}$ | $86.9_{\pm 0.6}$ | $\mathbf{88.8}_{\pm 0.5}$ | $0.58_{\pm 0.03}$ | $0.46_{\pm 0.02}$ | $0.42_{\pm 0.02}$ |
| TREC | 3 | $85.1_{\pm 0.9}$ | $90.7_{\pm 0.7}$ | $\mathbf{92.4}_{\pm 0.6}$ | $0.62_{\pm 0.04}$ | $0.48_{\pm 0.03}$ | $0.44_{\pm 0.02}$ |
| **6-Layer Model** | | | | | | | |
| SST-2 | 2 | $82.3_{\pm 0.7}$ | $90.4_{\pm 0.5}$ | $\mathbf{91.1}_{\pm 0.4}$ | $0.51_{\pm 0.03}$ | $0.39_{\pm 0.02}$ | $0.37_{\pm 0.02}$ |
| TREC | 3 | $84.9_{\pm 0.8}$ | $94.1_{\pm 0.5}$ | $\mathbf{94.7}_{\pm 0.4}$ | $0.57_{\pm 0.03}$ | $0.36_{\pm 0.02}$ | $0.34_{\pm 0.02}$ |

tasks. In addition to accuracy, HOBA consistently yields lower or comparable loss values: the 3-layer HOBA loss ranges from 0.36–0.69, compared to 0.28–0.90 in 2D. Moreover, HOBA achieves 10–48% lower KL loss than 2D by exhibiting more efficient knowledge distillation across datasets. Surprisingly, the 2D model performs slightly better than HOBA on the small TREC dataset (5.5K samples), due to the limited training size. Since HO attention is designed to capture complex token triplet interactions, its expressive capacity may be underutilized on small datasets. Additional qualitative visualizations on attention patterns and token interaction graphs are provided in **Appendix A.4**. Overall, HOBA provides efficient and robust modeling of token triplets within localized blocks, offering deeper contextual knowledge and outperforming standard 2D attention under more challenging training conditions.

Table 2: Performance comparison between HOBA and standard 2D attention models across five datasets. Results averaged over three independent runs.

| Dataset | Classes | Accuracy | | Total Loss | | KL Loss | |
|---------|---------|----------|-----|------------|-----|---------|-----|
| | | HOBA | 2D | HOBA | 2D | HOBA | 2D |
| **3-Layer Model** | | | | | | | |
| AG News | 4 | $\mathbf{92}_{\pm 0.5}\%$ | $92_{\pm 0.4}\%$ | $0.51_{\pm 0.02}$ | $0.28_{\pm 0.01}$ | $0.18_{\pm 0.01}$ | $0.20_{\pm 0.01}$ |
| MNLI | 3 | $\mathbf{64}_{\pm 0.7}\%$ | $62_{\pm 0.9}\%$ | $0.69_{\pm 0.02}$ | $0.61_{\pm 0.03}$ | $0.21_{\pm 0.01}$ | $0.30_{\pm 0.04}$ |
| SST | 2 | $\mathbf{80}_{\pm 0.8}\%$ | $79_{\pm 0.6}\%$ | $0.49_{\pm 0.03}$ | $0.90_{\pm 0.04}$ | $0.25_{\pm 0.08}$ | $0.24_{\pm 0.02}$ |
| TREC | 5 | $91_{\pm 0.6}\%$ | $\mathbf{94}_{\pm 0.5}\%$ | $0.39_{\pm 0.02}$ | $0.40_{\pm 0.05}$ | $0.14_{\pm 0.01}$ | $0.20_{\pm 0.01}$ |
| SQuAD | 4 | $\mathbf{97}_{\pm 0.4}\%$ | $93_{\pm 0.6}\%$ | $0.36_{\pm 0.02}$ | $0.56_{\pm 0.03}$ | $0.11_{\pm 0.01}$ | $0.21_{\pm 0.06}$ |
| **6-Layer Model** | | | | | | | |
| AG News | 4 | $\mathbf{93}_{\pm 0.6}\%$ | $92_{\pm 0.5}\%$ | $0.43_{\pm 0.03}$ | $0.36_{\pm 0.02}$ | $0.13_{\pm 0.01}$ | $0.18_{\pm 0.01}$ |
| MNLI | 3 | $\mathbf{77}_{\pm 0.6}\%$ | $76_{\pm 0.7}\%$ | $0.65_{\pm 0.03}$ | $0.98_{\pm 0.05}$ | $0.20_{\pm 0.02}$ | $0.33_{\pm 0.02}$ |
| SST | 2 | $\mathbf{81}_{\pm 0.5}\%$ | $80_{\pm 0.4}\%$ | $0.40_{\pm 0.02}$ | $0.45_{\pm 0.03}$ | $0.15_{\pm 0.01}$ | $0.23_{\pm 0.02}$ |
| TREC | 5 | $90_{\pm 0.7}\%$ | $\mathbf{95}_{\pm 0.5}\%$ | $0.51_{\pm 0.03}$ | $0.16_{\pm 0.01}$ | $0.15_{\pm 0.02}$ | $0.17_{\pm 0.01}$ |
| SQuAD | 4 | $\mathbf{98}_{\pm 0.3}\%$ | $95_{\pm 0.4}\%$ | $0.39_{\pm 0.02}$ | $0.76_{\pm 0.04}$ | $0.12_{\pm 0.01}$ | $0.18_{\pm 0.02}$ |

**Scratch Training Behavior.** We trained HOBA variants (3 & 6 layers ) from scratch without any distillation using same configuration. Representative results on AG-News (4 classes) and SST-2 (2 classes) are shown in Table 3. Based on the results, we find that scratch training is noticeably less stable in the early phase and requires substantially more warm-up to reach competitive performance. Basically, training without KD exhibits higher gradient noise and slower convergence during the first 8–12k updates, leading to final accuracies that are consistently 1–4% lower across seeds. Variance across runs is also meaningfully higher, making controlled comparisons with 2D baselines less reliable. These properties make KD the more appropriate choice for clean architectural comparisons over scratch training.

Table 3: Scratch (S) vs. KD training performance

**Comparative Evaluation with Recent 2D Attention.** To the best of our knowledge, no prior work has explored 3D attention mechanisms due to their high computational cost. Thus, we compare HOBA against recent 2D-based models, WIGRAPH-

| Model | AG-S | AG-KD | SST2-S | SST2-KD |
|-------|------|-------|--------|---------|
| HOBA (3L) | $89.1_{\pm 0.8}$ | $92.3_{\pm 0.3}$ | $77.4_{\pm 0.9}$ | $80.3_{\pm 0.4}$ |
| HOBA (6L) | $87.8_{\pm 0.9}$ | $93.6_{\pm 0.4}$ | $78.4_{\pm 1.1}$ | $81.8_{\pm 0.5}$ |

RoBERTa (Sekhon et al., 2023), TransEvolve-fullFF-1 (Dutta et al., 2021), and E2LN($R$) (Zhang

et al., 2024) which closely align with our task objectives. We ensure fair comparison by utilizing identical configurations for all models, training each for five epochs with sequence length 128 for AG News and 512 for IMDB. We find that HOBA reaches higher accuracy(see Table 4) in all cases by demonstrating the benefits of modeling triplet interactions. We exclude popular methods such as FlashAttention (Dao et al., 2022), Linformer (Wang et al., 2020) etc., from direct benchmarking, as they focus on kernel-level optimizations or non-attention operators. Our work instead targets structural improvements to attention via HO block interactions, making these methods complementary rather than directly comparable.

Table 4: Performance comparison between our HOBA model and recent 2D attention methods.

| Model | Dataset | Acc (Theirs/HOBA) | Their Focus | Their Interaction |
|---|---|---|---|---|
| WIGRAPH-RoBERTa (AAAI '23) | AG News | 91.52% / **95.22%** | Word interaction learning for 2D interaction | 2D |
| TransEvolve-fullFF-1 (NIPS '21) | AG News | 90.10% / **91.63%** | Parameter reduction for quadratic attention | 2D |
| E2LN(R) (ACL '24) | IMDB | 59.23% / **90.41%** | Efficient LayerNorm personalization | 2D |

**Comparison with Different Sparse Attention and Long-Context Benchmarks.** We test HOBA on long-sequence sentiment classification tasks using the IMDB, LRA-Text and Yelp Polarity datasets against several sparse mechanisms such as Longformer (Beltagy et al., 2020), BigBird (Zaheer et al., 2020), Local (Parmar et al., 2018), Block-diagonal(like ours approach), and Dilated attention (Hassani & Shi, 2022). These methods are widely adopted because their sparsity patterns reduce quadratic costs while preserving context. We choose these patterns for a fair comparison as they capture different trade-offs between locality, global context, and efficiency. The sparse baselines are configured with their typical hyperparameters. For instance, Longformer uses a sliding window of 512, BigBird combines 10% random + 10% global + 80% local links, Local attention uses a fixed window of 128, and Dilated attention uses a dilation rate of 2, ensuring settings align with our experimental context. As shown in Table 5, HOBA consistently achieves the best accuracy at $L$=512, reaching 93.1% on IMDB and 97.2% on Yelp, while also providing faster training than Longformer and Local attention.

Table 5: Comparison of HOBA (ours), Longformer, BigBird, and 2D attention baselines (2D Block-Diagonal, 2D Local, 2D Dilated; all without HO) on IMDB, Yelp Polarity, and LRA–Text across sequence lengths $L = 512$–$2048$. Results are reported as accuracy / loss / training time (minutes).

| Dataset | $L$ | HOBA (Ours) | Longformer | BigBird | 2D Block-Diagonal | 2D Local | 2D Dilated |
|---|---|---|---|---|---|---|---|
| | | Acc/Loss/Time | Acc/Loss/Time | Acc/Loss/Time | Acc/Loss/Time | Acc/Loss/Time | Acc/Loss/Time |
| IMDB | 512 | **93.1/0.23/10m** | 92.1/0.37/26m | 90.4/0.22/15m | 77.7/0.51/37m | 83.2/0.43/39m | 85.0/0.40/35m |
| | 2048 | – | 92.0/0.24/54m | 92.1/0.28/38m | – | – | – |
| Yelp Polarity | 512 | **97.2/0.09/280m** | 92.0/0.68/690m | 95.0/0.08/**166m** | 88.8/0.30/693m | 93.4/0.23/711m | 91.5/0.26/705m |
| | 1024 | **96.4/0.12/420m** | 92.1/0.70/1120m | 95.3/0.10/240m | – | – | – |
| | 2048 | **95.9/0.16/610m** | 93.2/0.61/1409m | 96.1/0.08/240m | – | – | – |
| LRA–Text | 1024 | **64.1/1.21/95m** | 63.8/1.27/210m | 65.2/1.19/160m | 57.8/1.40/260m | 60.3/1.32/275m | 62.1/1.29/250m |
| | 2048 | **63.4/1.25/180m** | 63.1/1.32/380m | 65.0/1.22/300m | – | – | – |

Across long context settings, HOBA remains stable and competitive even as sequence lengths increase to 1024 and 2048 tokens. On Yelp Polarity, HOBA consistently outperforms Longformer and remains close to BigBird even at 2k tokens, while maintaining significantly lower inference time. On the more challenging LRA–Text benchmark, HOBA again matches or exceeds Longformer and stays within a narrow margin of BigBird, despite using a simpler triplet formulation and without any specialized long-range architectural tuning. The main benefit comes from HOBA's ability to model richer token relationships using structured 3D attention over triplets. Unlike sparse models, which reduce complexity by restricting which tokens attend to each other, our HOBA model captures detailed local interactions through block-diagonal attention. This allows better context modeling without requiring long sequences or extensive computation. Therefore, even without support for sequences longer than 512, HOBA exceeds both Longformer and BigBird within their optimal settings(i.e., longer sequence length = 2048).

**Efficiency Gains with HOBA.** We sweep hidden size (model width) and plot accuracy against total FLOPs while holding all other hyperparameters fixed: 2 layers, 4 heads, FFN ratio $4H$, block size = 32, sequence length $L$=128, identical data, optimizer, epochs, and training schedule for a fair comparison of HOBA to standard 2D attention. FLOPs include attention, QKV/OUT projections, and FFN. For the smallest width setting ($H$=64), both HOBA and 2D RoBERTa have roughly 3.3M parameters, with memory usage of 13.2 MB for HOBA and 14.1 MB for 2D. The throughput is about 1.2k tokens/s for 2D and 1.6k tokens/s for HOBA. Figure 2(a) shows that **HOBA achieves better accuracy at lower compute across all widths**. Concretely, HOBA reduces FLOPs by $\approx 8-12\%$ relative to 2D (e.g., 33.6M$\rightarrow$30.2M at width 64; 337.6M$\rightarrow$304.0M at width 224) while improving accuracy by +1.5–3.3 points (e.g., $74.9\% \rightarrow 76.4\%$, $85.1\% \rightarrow 88.4\%$). Because projections and the FFN dominate at short contexts ($L$=128), the observed savings arise solely from the attention mechanism; the gap is expected to widen for longer sequences where attention costs dominate. The global Pareto frontier therefore shifts upward with HOBA, implying a better efficiency–performance trade-off than standard 2D attention at matched training conditions.

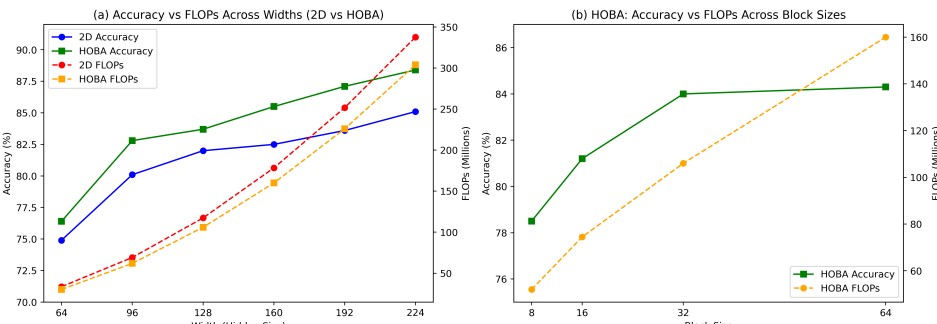

Figure 2: Computation vs. accuracy on AG News. (a) Accuracy and FLOPs across hidden widths for 2D vs. HOBA, showing that HOBA consistently achieves higher accuracy with lower compute. (b) Accuracy improves as block size increases, but saturates beyond $b$=32, identifying it as the best trade-off point.

**Generalization Bound via Rademacher Complexity.** We analyze why HOBA yields stronger generalization than standard 2D attention using Rademacher complexity (Bartlett & Mendelson, 2002; Mohri et al., 2018). For a loss $\ell$ bounded in $[0, 1]$ and $L_\ell$-Lipschitz, the standard bound states that with probability at least $1 - \delta$, for any $h \in \mathcal{H}$,

$$\mathcal{L}(h) \leq \hat{\mathcal{L}}(h) + 2L_\ell \hat{\mathfrak{R}}_n(\mathcal{H}) + 3\sqrt{\frac{\ln(2/\delta)}{2n}}. \qquad (9)$$

where $\hat{\mathfrak{R}}_n(\mathcal{H})$ is the empirical Rademacher complexity. Since $\mathcal{H}_{\text{HOBA}} \subseteq \mathcal{H}_{\text{Full}}$ by construction (block-diagonal masking strictly restricts token interactions), monotonicity of Rademacher complexity implies

$$\hat{\mathfrak{R}}_n(\mathcal{H}_{\text{HOBA}}) \leq \hat{\mathfrak{R}}_n(\mathcal{H}_{\text{Full}}). \qquad (10)$$

Therefore, whenever the empirical risk of HOBA is comparable to or lower than that of standard 2D attention, the reduced hypothesis class size guarantees a strictly tighter generalization bound. This progress stems from enforcing structured block-diagonal and overlapping attention masks, which limit the effective hypothesis space without sacrificing expressivity. Empirical evidence supports this theoretical claim (see Table 5, 6, 1, and 2). HOBA not only improves accuracy over 2D but also achieves 10–48% lower KL loss across datasets, confirming that the restricted hypothesis class translates into stronger generalization in practice (see **Appendix A.1** for the supporting lemma).

### 4.3 ABLATION STUDY

**Impact of Cross-Block Interaction.** Cross-block(CB) interaction in HOBA allows the model to connect information between distant token blocks. This link is especially useful when local attention alone cannot capture key semantic patterns. We find that enabling cross-block (CB) interaction consistently improves accuracy for both 2D and HOBA, but the gains are far more pronounced in

HOBA. On AG News and SST-2, HOBA with CB outperforms all other variants. The largest improvement is observed on MNLI, where HOBA with CB achieves 64% accuracy, surpassing both HOBA without CB (57%) and 2D with CB (53%). MNLI involves reasoning across sentence pairs; therefore, performance depends on connecting information that might sit far apart. Without CB interaction, those links break, and attention becomes limited. HOBA with CB forms triplets spanning the full sequence by allowing signals to combine across distant blocks. This strengthens reasoning, lessens noise, and promotes consistent patterns across layers. All these results highlight that while CB benefits both architectures, its combination with HOBA yields the most robust improvements, confirming HOBA + CB as the strongest configuration.

**Effect of Block Size.** At fixed context ($L$=128) and capacity (2 layers, width 128, 4 heads, FFN $4H$) increasing the HOBA block size from $b$=8, 16, 32, 64 steadily improves accuracy from 78.5% to 84.3%, while FLOPs grow roughly linearly in $b$ (52M to 160M)(see Figure 2(b)). Gains saturate beyond $b$=32 (only +0.3 points at $b$=64 for $\sim$1.5$\times$ more compute), identifying $b$=32 as the best accuracy–compute trade-off under this setting. We therefore suggest using $b$=32 unless longer-range within-block context is required.

Table 6: Ablation study comparing 2D (with and without CB) and HOBA variants (with and without CB). Results are accuracy/loss (block size 16, 3 layers).

| Dataset | Classes | 2D w/o CB | 2D w/ CB | HOBA w/ CB | HOBA w/o CB |
|---|---|---|---|---|---|
| AG News | 4 | 88% / 0.83 | 89% / 0.52 | **92% / 0.51** | 90% / 0.53 |
| SST-2 | 2 | 76% / 0.50 | 77% / 0.49 | **79% / 0.49** | 78% / 0.50 |
| MNLI | 3 | 51% / 0.73 | 53% / 0.71 | **64% / 0.69** | 57% / 0.73 |

**Effect of Sequence Length.** We conduct a sequence-length study to evaluate the robustness of HOBA under diverse input lengths across AG News, MNLI, and SST-2 datasets(see Table 9 in **Appendix A.5**). Results indicate that performance steadily improves as sequence length increases from 128 to 384. This trend is evident in tasks requiring deeper contextual understanding like MNLI, where accuracy increases from 51% to 59% and loss drops notably. This trend also reveals HOBA's capacity to capture broader dependencies with its HO structure. Even on SST-2, which has shorter inputs, increasing the sequence length to 256 increases accuracy. It also reduces loss, indicating that our model benefits from a longer context window.

**Controlled Comparison with a 2D Block-Local Baseline.** We construct a controlled experiment (block size $b = 32$, overlap $\lambda = 0.25$).) on AGNews comparing a 2D block-local attention model with HOBA (3D), using the same block size, overlap ratio, and cross-block routing. The only difference is that the 2D baseline performs pairwise (2D) attention, whereas HOBA performs triplet-wise (3D) attention. The results indicate that HOBA consistently outperforms the matched 2D baseline, demonstrating that the gains arise from HO (3D) attention rather than the block-local structure alone. The results indicate that HOBA consistently outperforms the matched 2D baseline, demonstrating that the gains arise from HO (3D) attention rather than the block-local structure alone.

## 5 CONCLUSION

This study proposes HOBA, a novel HO block-diagonal attention method that models token triplet interactions to improve contextual representation in transformer-based models. HOBA presents a 3D attention tensor while retaining computational efficiency through block-diagonal decomposition, unlike standard 2D attention, which uses pairwise token dependencies. Fur-

Table 7: 2D block-local attention vs. HOBA.

| Depth | Model | Acc. (%) | Loss |
|---|---|---|---|
| 3-layer | 2D Block | $88.6 \pm 0.3$ | $0.23 \pm 0.3$ |
| 3-layer | HOBA (3D) | $\mathbf{91.9 \pm 0.2}$ | $\mathbf{0.21 \pm 0.01}$ |
| 6-layer | 2D Block | $90.1 \pm 0.1$ | $0.34 \pm 0.04$ |
| 6-layer | HOBA (3D) | $\mathbf{93.3 \pm 0.2}$ | $\mathbf{0.34 \pm 0.02}$ |

thermore, we evaluate how cross-block interaction further enriches global reasoning without incurring substantial overhead. Extensive experiments across eight benchmark NLP datasets demonstrate that HOBA achieves competitive or superior accuracy with lower resource usage than vanilla or other sparse attention baselines. Our work enhances explainability by demonstrating interpretable token triplet interactions and how cross-block interaction helps the model to capture long-range semantic dependencies across disjoint token regions. In the future, we aim to explore adaptive HO attention mechanisms with dynamic block structures for NLP applications.

**Reproducibility Statement**

We ensure reproducibility by detailing our model architecture, training setup, and evaluation protocols in the main text, with additional information in Appendix. Our coding implementation is provided as an anonymized zipped package in the supplementary materials, including scripts for training, and evaluation.

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

# A APPENDIX

## A.1 PROOF OF LEMMA 1

**Lemma 1** (HOBA is a restricted sub-class of full attention). *Let $\mathcal{H}_{Full}$ be the hypothesis class induced by a transformer with unconstrained (dense) self-attention, and let $\mathcal{H}_{HOBA}$ denote the same architecture with block-diagonal or overlapping attention masks. Then $\mathcal{H}_{HOBA} \subseteq \mathcal{H}_{Full}$.*

**Setup.** Consider a single attention layer with per-head dimension $d$ and additive mask $M \in \mathbb{R}^{L \times L}$. The masked attention output is

$$\text{Attn}(Q, K, V; M) = \text{softmax}\left(\frac{1}{\sqrt{d}} QK^\top + M\right) V, \tag{11}$$

where $Q, K, V \in \mathbb{R}^{L \times d}$ are the query, key, and value projections. Let $f(X; \Theta, M)$ denote the model function with parameters $\Theta$ and attention mask $M$, consistent with the general $f_\theta(X)$ formulation in the main text. Then $f(x; \Theta, M)$ specifies a depth-$D$ transformer (with fixed depth and hidden size for both models) obtained by composing layers of the masked attention in Eq. (11).

**Classes.** Define the mask sets and induced hypothesis classes:

$$\mathcal{M}_{\text{Full}} := \mathbb{R}^{L \times L},$$

$$\mathcal{M}_{\text{HOBA}} := \left\{ M \in \mathbb{R}^{L \times L} : M_{ij} = -\infty \text{ if } (i,j) \text{ lies outside allowed block positions} \right\}, \quad (12)$$

$$\mathcal{H}_{\text{Full}} := \left\{ x \mapsto f(x; \Theta, M) : \Theta \text{ arbitrary}, M \in \mathcal{M}_{\text{Full}} \right\},$$

$$\mathcal{H}_{\text{HOBA}} := \left\{ x \mapsto f(x; \Theta, M) : \Theta \text{ arbitrary}, M \in \mathcal{M}_{\text{HOBA}} \right\}. \quad (13)$$

**Inclusion.** By construction, $\mathcal{M}_{\text{HOBA}} \subseteq \mathcal{M}_{\text{Full}}$, since block-diagonal or overlapping masks are a strict subset of all possible real-valued masks. Hence, for any $(\Theta, M)$ with $M \in \mathcal{M}_{\text{HOBA}}$, the same pair $(\Theta, M)$ is admissible in $\mathcal{H}_{\text{Full}}$. This implies $f(\cdot; \Theta, M) \in \mathcal{H}_{\text{Full}}$, and therefore every function realizable by HOBA is realizable by the full-attention model:

$$\mathcal{H}_{\text{HOBA}} \subseteq \mathcal{H}_{\text{Full}}.$$

The argument extends directly to multi-head and multi-layer transformers, since masks are applied independently per head and per layer, and compositionality preserves inclusion. $\qquad\square$

### A.2 MORE ON DATASETS

The datasets used in our experiments span multiple NLP tasks, including news categorization (AG News), natural language inference (MNLI), sentiment analysis (SST, IMDB, Yelp Polarity), question classification (TREC), and question answering (SQuAD). They vary widely in both scale and input length: AG News and MNLI contain medium-length sequences with median sizes of about 35–40 tokens, SST and TREC are shorter with median lengths under 20 and 10 tokens respectively, while SQuAD passages are longer with a median around 150 tokens. IMDB and Yelp Polarity consist of full-length user reviews, exhibiting much greater input size, with medians of approximately 230 and 150 tokens. This diversity in task type, size, and sequence length provides a comprehensive benchmark for evaluating both efficiency and effectiveness of attention mechanisms.

Table 8: Dataset Statistics and Task Descriptions

| Dataset | Task | Classes | Train | Test |
|---------|------|---------|-------|------|
| AG News | NC | 4 | 120K | 7.6K |
| MNLI | NLI | 3 | 393K | 9.8K |
| SST | SA | 2 | 67K | 1.8K |
| TREC | QC | 5 | 5.5K | 500 |
| SQuAD | QA | 4 | 87K | 10.6K |
| IMDB | SA | 2 | 25K | 25K |
| LRA–Text | SA | 2 | 250K | 25K |
| Yelp Polarity | SA | 2 | 260K | 20K |

### A.3 BLOCK-WISE TENSOR ATTENTION

Our HOBA takes an input sequence $X \in \mathbb{R}^{L \times d}$ and projects it into query, key, and value subspaces as $Q, K_1, K_2, V_1, V_2 \in \mathbb{R}^{L \times d}$ through learned linear transformations (Algorithm 1). The sequence is divided into overlapping blocks $\mathcal{B}$ of size $b$ with overlap ratio $\lambda$, and for each block $(s, e) \in \mathcal{B}$ we extract the block tensors $Q^{(m)}, K_1^{(m)}, K_2^{(m)}, V_1^{(m)}, V_2^{(m)} \in \mathbb{R}^{b \times d}$.

**Higher-order interaction tensor.** Within each block, the 3D attention interaction is computed using the tensor contraction

$$\mathcal{T}^{(m)} = \text{einsum}(\texttt{"ijd,jkd,jld->ijkl"}, Q^{(m)}, K_1^{(m)}, K_2^{(m)}) \in \mathbb{R}^{b \times b \times b},$$

where $i$ indexes the query token, $(j, k)$ index paired key tokens, and the embedding dimension $d$ is contracted out. Softmax over the $(j, k)$ axes yields the triplet attention tensor $\mathcal{A}^{(m)} \in \mathbb{R}^{b \times b \times b}$.

**Value interaction and block output.** The corresponding value interaction tensor is

$$\mathcal{U}^{(m)} = \text{einsum}(\texttt{"jkd, jld->jkld"}, V_1^{(m)}, V_2^{(m)}) \in \mathbb{R}^{b \times b \times d}.$$

The contextual block output is then obtained by

$$C^{(m)} = \text{einsum}(\texttt{"ijkl, jkld->id"}, \mathcal{A}^{(m)}, \mathcal{U}^{(m)}) \in \mathbb{R}^{b \times d},$$

which provides one contextual representation per token inside the block.

**Aggregation and multi-head computation.** All $C^{(m)}$ outputs are accumulated into the global tensor $O$ with overlap handling. In the multi-head setting, the same computation is applied independently for each head $h$ to obtain $C_h^{(m)}$, after which the outputs are concatenated across heads and passed through the usual output projection $W_O$. A pooled summary representation $R$ is obtained via BLOCKPOOLING and incorporated through CROSSBLOCKATTENTION, yielding the final block-updated representation.

---

**Algorithm 1** Block-Diagonal Higher-Order Attention

---

**Require:** Input $X \in \mathbb{R}^{L \times d}$, block size $b$, overlap ratio $\lambda$
**Ensure:** Output $O \in \mathbb{R}^{L \times d}$
1: *// Project to attention subspaces*
2: $Q, K_1, K_2, V_1, V_2 \leftarrow XW_Q, XW_{K_1}, XW_{K_2}, XW_{V_1}, XW_{V_2}$
3: *// Initialize output and generate overlapping blocks*
4: $O \leftarrow 0_{L \times d}$, $\mathcal{B} \leftarrow$ GENERATEBLOCKS$(L, b, \lambda)$
5: **for** each block $(s, e) \in \mathcal{B}$ **do**
6:     *// Extract block tensors*
7:     $Q^{(m)}, K_1^{(m)}, K_2^{(m)}, V_1^{(m)}, V_2^{(m)} \leftarrow X[s : e]$
8:     *// Compute 3D attention tensor*
9:     $\mathcal{T}^{(m)} \leftarrow$ EINSUM$(\texttt{"ijd, jkd, jld->ijkl"}, Q^{(m)}, K_1^{(m)}, K_2^{(m)})/\sqrt{d}$
10:    $\mathcal{A}^{(m)} \leftarrow$ SOFTMAX$(\mathcal{T}^{(m)})$
11:    *// Value interaction tensor*
12:    $\mathcal{U}^{(m)} \leftarrow$ EINSUM$(\texttt{"jkd, jld->jkld"}, V_1^{(m)}, V_2^{(m)})$
13:    *// Contextual block output*
14:    $C^{(m)} \leftarrow$ EINSUM$(\texttt{"ijkl, jkld->id"}, \mathcal{A}^{(m)}, \mathcal{U}^{(m)})$
15:    *// Accumulate with overlap*
16:    $O[s : e] \leftarrow O[s : e] + C^{(m)}$
17: **end for**
18: *// Cross-block interaction*
19: $R \leftarrow$ BLOCKPOOLING$(O, \mathcal{B})$
20: $O \leftarrow O +$ CROSSBLOCKATTENTION$(O, R)$
21: *// Normalize overlaps and project*
22: $O \leftarrow$ NORMALIZEOVERLAPS$(O)W_O$
23: **return** $O$

---

### A.4 COMPARISON OF TOKEN-LEVEL ATTENTION PATTERNS

We visualize the interaction graphs generated by HOBA and standard 2D models to understand the qualitative differences in token-level attention patterns. Comparison of attention patterns shown in Figure 3 and Figure 4 analogizes token-level attention interactions between a standard self-attention model (left) and the proposed HOBA mechanism (right). Both graphs visualize attention among tokens in the sentence "Global markets showed uncertainty amid economic volatility." Edge thickness and color express the relative strength of attention (red = strongest, gray/green = weakest). The softmax-based model displays dense, diffused relations with limited focus, where many token pairs acquire comparable attention. In contrast, HOBA shows a structured pattern: fewer but sharper edges concentrate around key semantic units (e.g., "uncertainty" $\leftrightarrow$ [CLS], "Global" $\leftrightarrow$ "volatility"), reflecting its HO block-localized computation. This visualization reveals HOBA's ability to highlight salient interactions while filtering irrelevant ones, enhancing interpretability and modeling focus.

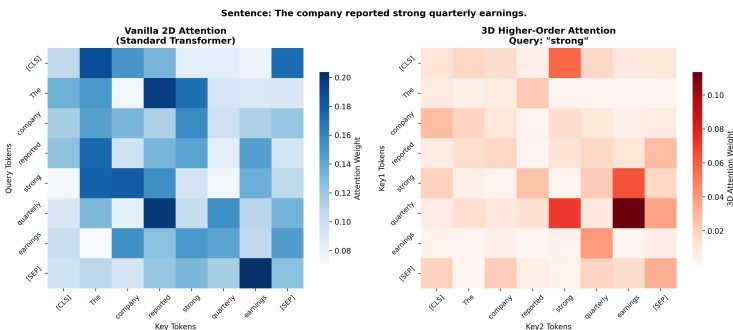

Figure 3: Comparison of attention patterns between standard 2D attention (left) and our proposed 3D higher-order attention (right). The 2D plot displays traditional pairwise query-key token interactions, while the 3D plot shows a $key_1 \times key_2$ slice for the query token *"strong"* by capturing triplet interactions. HOBA assigns a higher weight to semantically relevant triplets such as (*"strong," "quarterly,"* and *"earnings"*) and reflects its ability to model richer contextual dependencies.

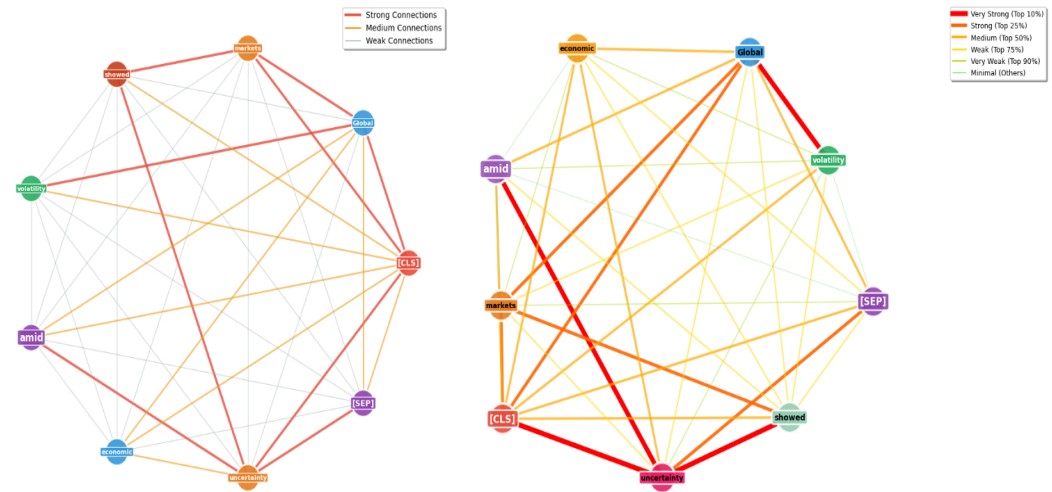

Figure 4: **Comparison of Token Interaction Patterns.** Left: Traditional 2D self-attention shows widespread but diluted connections across tokens. Right: HOBA generates sparse, structured attention with strong localized dependencies. Token pairs such as ("uncertainty", "[CLS]") and ("Global", "volatility") receive top-rank attention in HOBA, demonstrating its ability to capture critical HO semantics with sharper focus.

## A.5    EFFECT OF SEQUENCE LENGTH ABLATION

Table 9: Sequence length ablation for HOBA (block=32, 6 layers). Accuracy / loss shown.

| Dataset | L=128 | L=256 | L=384 |
|---------|-------|-------|-------|
| AG News | 90% / 0.46 | 91% / 0.48 | **93% / 0.43** |
| MNLI | 51% / 0.72 | 51% / 0.69 | **59% / 0.53** |
| SST-2 | 77% / 0.43 | **78% / 0.34** | N/A |

## A.6    ACCOUNTING FOR OVERLAP AND CROSS-BLOCK COSTS

This section provides the full derivation of the additional computational terms introduced by overlapping blocks (Eq. 4) and cross-block attention (Eqs. 5–7), complementing the complexity discussion in Section 3.2.

**(a) Overlap cost (referencing Eq. 4).** From Eq. 4, the final token output is obtained by averaging the block-specific outputs $o_i^{(m)}$ over the set of overlapping blocks $\mathcal{B}_i$ that contain token $x_i$:

$$o_i = \frac{1}{|\mathcal{B}_i|} \sum_{m \in \mathcal{B}_i} o_i^{(m)}.$$

Since each token appears in a constant number of overlapping blocks determined by the fixed overlap ratio $\lambda$, this introduces only a constant-factor overhead and therefore does not change the asymptotic complexity of the model.

**(b) Cross-block attention cost (referencing Eqs. 5–7).** As defined in Eqs. 5–7, each block summary representation $r^{(m)}$ is computed by pooling over its $b$ token-level outputs:

$$r^{(m)} = \frac{1}{b} \sum_{i=mb}^{(m+1)b-1} h_i,$$

where $h_i$ denotes the intermediate contextual output for token $x_i$. Cross-block attention is then performed across all $B = L/b$ blocks. The attention operation over these $B$ block representatives requires

$$\mathcal{O}(B^2 d) = \mathcal{O}\left(\frac{L^2}{b^2}d\right),$$

which captures the full cost of global block-to-block interactions.

**(c) Final combined complexity.** Combining the local 3D block cost $\mathcal{O}(Lb^2 d)$ with the overlap (constant factor) and the cross-block attention term above, the overall complexity of HOBA is

$$\text{HOBA}(L) = \mathcal{O}(Lb^2 d) + \mathcal{O}\left(\frac{L^2}{b^2}d\right),$$

which correctly includes all contributions from local higher-order interactions, overlap aggregation, and cross-block attention.

## A.7 Key Difference with SOTA Methods

We find that all these prior higher-order attention models (Table 10) operate globally (spectral kernels, full-sequence graphs, dense tensor contractions), resulting in fundamentally different computational objectives and scaling regimes. We find that their complexity is much higher than HOBA's block-local 3D attention, making direct comparison neither methodologically fair nor computationally feasible under matched FLOPs. However, Sparse-attention models (Longformer, BigBird, etc.) are the appropriate baselines because, like our HOBA, they are specifically designed to reduce the quadratic cost of self-attention while preserving long-range dependencies. They target the same efficiency–accuracy trade-off, operate under similar computational budgets, and scale to long sequences. Thus, it makes them conceptually and empirically aligned with HOBA's goals.

Table 10: Conceptual Comparison of HO Attention Methods and How HOBA Differs.

| Method | Interaction | How HO is Modeled | Comp. Structure | Limitation | How HOBA Differs |
|---|---|---|---|---|---|
| **Jump Attention (JAT)** (Zhou et al., 2022) | Spectral interactions | FFT-based spectral convs for multi-token relations | Global conv kernels | No explicit triplets; heavy compute | HOBA computes *explicit* $((i, j, k))$ triplets but only inside blocks |
| **WIGRAPH** (Sekhon et al., 2023) | Graph-based HO | Graph propagation for global relations | Global graph layer | Needs full-sequence graph; not scalable | HOBA uses local 3D blocks + lightweight cross-block routing |
| **Graph-Induced Attn** (Hong et al., 2022) | Graph-biased dot-prod | Injects syntactic graph into 2D attention | Standard attention + graph bias | Still pairwise; not 3D | HOBA directly models triplets; no external graph needed |
| **Kernelized HO Attn** (Choromanski et al., 2020) | Kernel approx. | Tensor kernels for HO interactions | Sequence-wide tensor ops | High memory, expensive | HOBA uses block-diagonal factorization; linear in $L$ for fixed block size |
| **Sparse HO Self-Attn** (Cui et al., 2019) | Sparse structured | Sparse softmax patterns | Standard 2D attention | No triadic modeling | HOBA performs true 3D attention over triplets within blocks |

