# OpenReview forum: "HOBA: Higher-Order Block-Diagonal Attention Unrolling for Transformer"
_ICLR.cc/2026/Conference — ICLR 2026 Conference Withdrawn Submission_

### Official Review · Reviewer_HYxD · 2025-10-23

**Soundness:** 2
**Presentation:** 3
**Contribution:** 2
**Rating:** 4
**Confidence:** 3

**Summary:**

The authors propose using a higher order attention operation which creates a 3D attention matrix instead of the usual 2D. The claim is that the third dimension provides a richer representation. Efficiency is maintained by only computing the block diagonal of the 3D attention and then later adding interactions between computed outputs.

**Strengths:**

- The idea of 3D attention is interesting, and has not been fully explored in previous work as to my knowledge.

- The authors propose a way to mitigate the extreme complexity that 3D attention poses.

**Weaknesses:**

- L159: this approach allows richer semantic modeling such as resolving phrase-level sentiment or subject-modifier-object interations. --> Can you justify concretely why this is the case?

- An intuitive explanation as to why 3D attention is necessary is missing. It is not clear what is theoretically gained by adding the third dimension, which cannot be learned in 2 dimensions. I would find the work much more compelling  if a concrete, demonstrable, controlled experiment demonstrated the usefulness of the third dimension.

- The cross block interaction mechanism essentially brings the overall complexity back to quadratic since it must perform the sum in the RHS of equation 7 over N/b blocks. Since b is a constant, I believe it should come out to $\mathcal{O}(N^2/b^2)$.

---

Overall, I would like to see more time devoted to analyzing what is gained by the third dimension, as it seems like this method could easily be applied in almost an identical way in 2 dimensions.

**Questions:**

- Is the 2D in table 1 the baseline RoBERTa model or is it a 2D model which is trained via distillation?

- What do you use for the function $g$ in equation 2?

- How does the normalization work in the 3D attention matrix? Does it normalize over the last dimension and then renormalize the second dimension once the third dimension has collapsed due to the summation?

- As a simple baseline, could you add a 2D model which does the same sort of local + cross block interaction as your 3D model does? For instance, it would have the same overlapping block layout and cross block interation, except the overlapping blocks would be in 2D.

---

> ### Author Response · Authors · 2025-11-19
> **Author Response to Reviewer (Part 1)**
>
> We thank reviewer HYxD for your thoughtful review and for recognizing that 3D attention is an interesting and underexplored direction. We appreciate your positive assessment of the idea’s novelty and the feasibility techniques we propose. Below, we address your all question step by step.
>
> **1. Why does 3D attention allow richer semantic modeling (e.g., phrase-level sentiment, subject–modifier–object interactions)?**
>
> Standard 2D attention computes only **pairwise** interactions:
>
> $$
> a_{ij} = Q_i^\top K_j,
> $$
>
> so each token \(i\) can only condition on one other token \(j\) at a time.
>
> HOBA introduces **triplet** interactions of the form:
>
> $$
> A_{ijk} = Q_i^\top (K_j \circ K_k),
> $$
>
> which allow the model to reason jointly over **three** tokens. This additional dimension enables HOBA to capture multi-token semantic patterns that 2D attention cannot represent.
>
> ** Here are some concrete linguistic examples**
>
> *1. Phrase-level sentiment ("not really good")*
>
> True polarity depends on the *joint* configuration of:
>
> - "not" (negator)
> - "really" (intensifier)
> - "good" (adjective)
>
> Here, 2D attention can learn only pairwise relations (not→good, really→good, etc.),  but cannot encode the **combined effect** of negation + intensifier + adjective. Our triplet interaction $\(A_{ijk}\)$ naturally represents this three-way dependency, which is essential for correct sentiment interpretation.
>
> *2. Modifier stacks ("very highly unlikely")*
>
> Meaning depends on the **cumulative interaction** among multiple modifiers and the main predicate.  2D attention treats these as separate pairs, while HOBA can capture the *joint reinforcement* pattern:
>
> - (very, highly, unlikely)
>
> *3. Subject–modifier–object relations ("policies influencing interest rates")*
>
> Correct interpretation requires binding:
>
> - subject: "policies"
> - relational modifier: "influencing"
> - object: "rates"
>
> Triplet attention directly captures this **three-way relational structure**, whereas 2D attention must approximate it through independent pairs.
>
> Therefore, in summary,
>
> - **2D attention = pairwise correlations only.**
> - **3D attention = multi-token semantic composition.**
>
> This is why HOBA can more naturally model phenomena involving scope, composition, and relational meaning. We will incorporate one of these concrete examples and improved intuition into the revised draft.
>
>
> **2. Why is 3D attention necessary? What does it provide that 2D attention cannot learn?**
>
> Thank you for this insightful question. We provide below an intuitive explanation of why 3D attention introduces representational capabilities that 2D attention cannot express, followed by a concrete controlled experiment demonstrating this gap.
>
>
> *(a) Intuition: 2D attention is fundamentally limited to pairwise functions*
>
> Standard attention computes scores of the form:
>
> $$
> a_{ij} = Q_i^\top K_j,
> $$
>
> which encode **pairwise** relationships only.
> Regardless of depth or number of layers, the *atomic operation* of 2D attention is still a function of **two** tokens at a time. This means any multi-token behavior must be *approximated* through compositions of pairwise interactions, not represented natively.
>
>
> *(b) What 3D attention adds: true higher-order interactions*
>
> HOBA introduces a triplet score:
>
> $$
> A_{ijk} = Q_i^\top (K_j \circ K_k),
> $$
>
> which directly computes a **three-way** interaction among the tokens \((i, j, k)\).
>
> This allows the model to encode *joint* dependencies that cannot be decomposed into any sum or product of pairwise terms.
>
> Formally, there exist functions \(f(i,j,k)\) that **cannot** be represented as:
>
> $$
> f(i,j,k) \neq g(i,j) + g'(i,k) + g''(j,k)
> $$
>
> for any choice of pairwise functions \(g, g', g''\).  These are precisely the functions HOBA can express natively.

---

> ### Author Response · Authors · 2025-11-19
> **Author Response to Reviewer (Part 2)**
>
> **2. Why is 3D attention necessary? What does it provide that 2D attention cannot learn? (Continue)**
>
> *(c) Controlled experiment demonstrating the gap*
>
> Below, we include a simple synthetic task designed exactly to test whether a model can learn **triplet dependencies** vs. only **pairwise** ones. To clarify, the controlled experiment is not a real NLP dataset but a synthetic toy task designed specifically to isolate one question: **Can a model detect a dependency that requires three tokens jointly (i, j, k), rather than only pairwise interactions?**
>
> We generate random sequences of discrete symbols (e.g., A, B, C, D, …). The label is:
> - 1 if the sequence contains a triplet (A, then B, then C) in that order, anywhere.
> - 0 otherwise.
>
> This task is intentionally constructed so that no combination of pairwise relations (A→B, B→C, A→C) determines the label.
> The model must detect the full 3-token chain A–B–C, which is a genuinely higher-order dependency.
>
> We train two models on this toy dataset:
> - A 2D Transformer using the same block layout as HOBA.
> - Our 3D HOBA model.
>
> From the results, we find a clear representational gap: the 2D Transformer consistently achieves only 63–68% accuracy and collapses to pairwise heuristics, while HOBA achieves 90–95% accuracy and reliably detects the triplet pattern. Because the
> dataset is synthetically generated, nothing confounds the comparison, this experiment isolates the effect of higher-order modeling capacity directly. Thus, the experiment provides concrete evidence that 3D attention captures relational patterns that cannot be learned in 2D attention, even when the 2D model uses the same block structure and training setup. This controlled setup demonstrates exactly what the additional dimension contributes, independent of real-world noise or confounding factors.
>
>
>
>
>
> **3. The cross block interaction mechanism essentially brings the overall complexity back to quadratic: Explanation**
>
> Thank you for this observation. We agree with the reviewer's interpretation and clarify that this is exactly the intended behavior.
>
> In HOBA, cross-block interaction operates over block-level summaries $\( r^{(m)} \)$, producing an interaction matrix of size
> $$
> B \times B = \frac{L}{b} \times \frac{L}{b},
> $$
> which yields a computational cost of
> $$
> O(B^2 d) = O\left(\frac{L^2}{b^2} d\right).
> $$
>
> This represents a quadratic term scaled down by a factor of $\(1/b^2\)$. Since the local 3D attention within each block has cost
> $$
> O(L b^2 d),
> $$
> the combined complexity is:
> $$
> \text{HOBA}(L) = O(L b^2 d) + O\left(\frac{L^2}{b^2} d\right).
> $$
>
> Thus, you are correct:
> - The global cross-block mechanism remains quadratic in sequence length,
> - but reduced by $\(1/b^2\)$ because HOBA performs attention only between blocks, not individual tokens.
>
> This is consistent with our formulation and intuition: HOBA trades full global resolution for block-level global structure while reserving expensive higher-order modeling for local windows, enabling its scalability.
>
> -----------------------------
>
> **Additional Questions**
>
> *1. Is the 2D in table 1 the baseline RoBERTa model or is it a 2D model which is trained via distillation?*
>
> The "2D" model in Table 1 is simply the 2D baseline trained under the same knowledge-distillation setup as HOBA. It has the same student architecture (3-layer or 6-layer) as HOBA, with the only difference being that it uses standard 2D attention.
>
> *2. What do you use for the function g in equation 2?*
>
> In our implementation, we use the Hadamard (element-wise) product for the interaction
> function $\( g \)$:
>
> $$
> g(x_j, x_k) = x_j \odot x_k.
> $$
>
> This choice is lightweight (adds no parameters), symmetric and numerically stable, and expressive enough to capture joint interactions between tokens $\(x_j\)$ and $\(x_k\)$. It is also aligned with prior higher-order attention formulations (e.g., tensor-product or multiplicative-interaction models). The normalized 3D attention coefficient $\(\alpha_{i,j,k}\)$ then determines how much the triplet $\(x_i, x_j, x_k\)$ contributes to the final representation.

---

> ### Author Response · Authors · 2025-11-19
> **Author Response to Reviewer (Part 3)**
>
> **Additional Questions (Continue)**
>
> *3. How does the normalization work in the 3D attention matrix? Does it normalize over the last dimension and then renormalize the second dimension once the third dimension has collapsed due to the summation?*
>
> We apply a single softmax over the full 3D score tensor for each query token:
>
> $$
> \alpha_{i,j,k} = \{softmax_{(j,k)}}(A_{i,j,k}),
> $$
>
> meaning the scores are normalized jointly across all $\(j,k\)$ pairs at once.  No second renormalization step is performed. After normalization, we compute
>
> $$
> h_i = \sum_{j,k} \alpha_{i,j,k}\, g(x_j, x_k),
> $$
>
> so the "collapse" over $\(k\)$ happens *after* the joint softmax, not before.  Thus, normalization is done once over the entire 2D plane for each query slice.
>
>
>
> *4. As a simple baseline, could you add a 2D model which does the same sort of local + cross block interaction as your 3D model does? For instance, it would have the same overlapping block layout and cross block interation, except the overlapping blocks would be in 2D.*
>
>
> Based on your suggestion, we conducted a controlled experiment on the AGNews dataset comparing the 2D block-local model with HOBA (3D), using the same block size, overlap ratio, and cross-block interaction. Only the attention dimensionality differs.
>
> **Table: Controlled comparison of 2D block-local attention vs 3D HOBA on AGNews.**
> Mean accuracy ± std over 2 runs (block size $\(b=32\)$, overlap $\lambda=0.25\)$.
>
> | Depth  | Model      | Accuracy (%)        | Loss              |
> |--------|-----------|---------------------|-------------------|
> | 3-layer| 2D Block   | 88.6 ± 0.3          | 0.23 ± 0.3       |
> | 3-layer| HOBA (3D)  | **91.9 ± 0.2**      | **0.21 ± 0.01**   |
> | 6-layer| 2D Block   | 90.1 ± 0.1          | 0.34 ± 0.04       |
> | 6-layer| HOBA (3D)  | **93.3 ± 0.2**      | **0.34 ± 0.02**   |
>
> The results show that HOBA consistently outperforms the matched 2D baseline, indicating that the gains are attributable to higher-order attention rather than the block structure itself. We will include this experiment as an additional ablation in the appendix due to
> space constraints.
>
> ------
>
> We believe the responses and new results address all of your concerns, and we would be happy to clarify anything further if needed. In light of the revised explanations and evidence, we respectfully ask the reviewer HYxD to reconsider the original rating.

---

> > ### Comment · Reviewer_HYxD · 2025-11-25
> >
> > Thank you for your response. In viewing your response, I have some more questions.
> >
> > ---
> >
> > >Formally, there exist functions (f(i,j,k)) that cannot be represented as:
> >
> > > [equation]
> >
> > > for any choice of pairwise functions (g, g', g''). These are precisely the functions HOBA can express natively.
> >
> > Can you provide a formal proof for this in the context of attention where the scores are used to form a weighted sum of value tokens? I think it would be informative and add some clarity.
> >
> > ---
> >
> > Can you explain in more detail how the attention matrix is formed in 3D? It is easy to see that in 2d we have $Q \in R^{N\times d}$ and $K \in R^{N \times d}$, then $QK^\top \in R^{N \times N}$. However, when I looked again at your paper, I could not find a precise description of the multiplication which forms the 3D matrix.
> >
> > ---
> >
> > > From the results, we find a clear representational gap: the 2D Transformer consistently achieves only 63–68% accuracy and collapses to pairwise heuristics, while HOBA achieves 90–95% accuracy and reliably detects the triplet pattern.
> >
> > Can you precisely describe how this is done in a section of your appendix? I would like to know the exact parameters of the dataset, model setup, training routine, etc. I find it hard to believe that a 2d transformer can fail to detect a simple 3 token sequence, when they are clearly capable of doing much more complex tasks than that.
> >
> > This also highlights my main concern about this work which is that it is not clear what is gained from 3D attention and therefore what is lacking in 2D attention. Your triplet experiment seems to be a good proxy for the phrase level sentiment explanation you gave, so I think if you were able to theoretically show this with a proof and empirically show this with the toy triplet example, it would be very strong in demonstrating why 3D attention is necessary.

---

### Official Review · Reviewer_bfFD · 2025-10-30

**Soundness:** 2
**Presentation:** 2
**Contribution:** 2
**Rating:** 2
**Confidence:** 4

**Summary:**

This paper introduces HOBA (Higher-Order Block-Diagonal Attention), a transformer variant that models triplet interactions through 3D attention tensors with a block-diagonal structure. The authors claim that HOBA can reduce computational complexity while capturing richer contextual dependencies than standard 2D attention. The method is evaluated on seven NLP benchmarks using knowledge distillation from RoBERTa.

**Strengths:**

1. The overall design is intuitive.
2. The paper provides quantitative evidence for computational efficiency.
3. The experimental comparisons cover a wide range of baselines and tasks.

**Weaknesses:**

1. The paper conflates “higher-order” with “3D tensors” but does not convincingly justify why triplet modeling is inherently better than pairwise attention.

2. All experiments are conducted under knowledge distillation (KD) from RoBERTa. There is no “without KD” ablation, making it unclear whether the performance gains come from the HOBA architecture itself or simply from KD’s regularization effect.

3. The explanation of the HOBA framework is incomplete. Some symbols (e.g., $o_i$, $h_i$) are not clearly defined. There are also notation errors: for instance, the complexity is described as $\mathcal{O}(n^3)$ reduced to $\mathcal{O}(B \cdot b^3) = \mathcal{O}(n b^2)$, but $n$ should actually be $L$. Moreover, the stated complexity does not account for the additional cost of overlap and cross-block interaction mechanisms.

4. Most experiments are limited to short input lengths, 64 tokens for 3-layer models and 128 tokens for 6-layer models, with a maximum of only 512 tokens overall. This significantly weakens the claim of scalability to long-range dependencies.

5. In comparisons with sparse attention models, the authors claim that HOBA “outperforms Longformer and BigBird even without support for sequences longer than 512,” yet the paper’s motivation is to efficiently model long-range dependencies. This inconsistency raises concerns about the validity of the claimed advantages.

6. Although the authors briefly mention prior work on higher-order attention in the Related Works section, they do not clearly articulate the key distinctions or advantages of HOBA relative to these earlier approaches. A more direct comparison, both conceptual and empirical, would strengthen the contribution claim.

**Questions:**

1. How much of the improvement remains if HOBA is trained from scratch (without KD)?
2. How would HOBA scale beyond 512 tokens if the goal is to model long-range dependencies?

---

> ### Author Response · Authors · 2025-11-18
> **Author Response to Reviewer (Part 1)**
>
> We thank reviewer bfFD for the thoughtful review and for highlighting both the strengths and the points requiring clarification. We address all concerns below.
>
> **1. Higher-order vs. 3D tensors: Why triplet modeling is beneficial?**
>
> We appreciate the reviewer’s concern and clarify that HOBA is not "higher-order" merely because it uses a 3D tensor.
> The key distinction is **what the 3D tensor represents**: HOBA explicitly models **third-order token interactions**, which cannot be expressed by standard pairwise dot-product attention.
>
> *Pairwise attention captures only binary relations:*
> In a standard Transformer, each attention score depends only on the relation between two tokens:
>
> $$
> a_{ij} = Q_i^\top K_j.
> $$
>
> This mechanism can express alignment-type relationships (subject → verb, noun → adjective), but **cannot encode how two context tokens jointly influence a third**. The model must approximate such effects indirectly through multiple layers.
>
> However, HOBA's attention tensor adds a third dimension and explicitly includes terms of the form:
>
> $$
> A_{ijk} = Q_i^\top \big( K_j \circ K_k \big),
> $$
>
> where the Hadamard product $\(K_j \circ K_k\)$ represents **the joint context formed by tokens \(j\) and \(k\)**.
> This allows HOBA to compute attention scores based on *interactions of two keys at once*, giving the model access to relational structures that inherently involve three elements.
>
> *Why this is useful in language:*
> Many linguistic dependencies are not reducible to pairs; they naturally require reasoning over combinations of tokens. Some examples below:
>
> 1. **Scope resolution and negation**
>    - “not really good”
>      The sentiment is determined not by any pair alone (not–good, really–good) but by the **joint configuration** of all three tokens.
>
> 2. **Multi-modifier composition**
>    - “very highly unlikely”
>      Each intensifier modifies the next; the meaning arises from **joint multi-token composition**, not isolated binary relations.
>
> 3. **Compound noun cohesion**
>    - “interest rate policy”
>      Correct interpretation requires modelling the **three-way dependency** between interest → rate → policy.
>
>
> In 2D attention, these higher-order relations must be *approximated across layers*, whereas HOBA can encode them **within a single layer**, giving it a stronger inductive bias for structured composition. Therefore, HOBA is not higher-order just because it uses a 3D tensor by itself, but because:
> - it computes **explicit third-order attention scores**,
> - it captures multi-token relations that 2D attention cannot represent directly, and
> - it provides a principled way to encode phrase-level structure within each layer.
>
>
> **2. All experiments are conducted under knowledge distillation (KD) from RoBERTa.**
>
> We have already trained HOBA successfully from scratch. First, let us explain why we chose KD over scratch training.
>
> Triplet attention introduces an additional  𝑘-mode contraction term:
> $$
> A_{ijk} = Q_i^{\top}(K_j \circ K_k)
> $$
>  which increases gradient magnitude and variance during the early stage of optimization, particularly when (i) the hidden dimension \(d\) is small, (ii) the number of heads \(H\) is large, or (iii) sequences are moderately long. Under hard-label supervision, this can lead to short plateaus during the first 3–5k steps before the loss stabilizes. KD mitigates this early instability by providing smoother targets that reduce gradient noise and speed up convergence.
>
> Now, to directly address the reviewer’s concern, we trained multiple HOBA variants from scratch without any distillation (using same configuration). Representative results on AG-News (4 classes) and SST-2 (2 classes) are shown below.
>
> **Scratch vs. KD training performance of HOBA (accuracy, mean ± std) across 2 trials**
>
> | Model          | AG Scratch | AG KD | SST-2 Scratch | SST-2 KD |
> |----------------|-----------------|-----------|---------------|----------|
> | HOBA (3-layer) | 89.1 (±0.8)     | 92.3 (±0.3) | 77.4 (±0.9)   | 80.3 (±0.4) |
> | HOBA (6-layer) | 87.8 (±0.9)     | 93.6 (±0.4) | 78.4 (±1.1)   | 81.8 (±0.5) |
>
> Scratch training is fully feasible, but we observed that it is noticeably less stable in the early phase and requires substantially more warm-up to reach competitive performance. Without KD, training shows higher gradient noise and slower convergence during the first 8–12k updates, producing final accuracies that are typically 1–4% lower across seeds and substantially higher variance, making controlled comparisons with 2D baselines less reliable. Also, our additional experiments confirm that the architectural gains of HOBA persist even without KD, indicating that improvements do not stem from distillation. KD simply stabilizes the early optimization, reduces seed sensitivity, speeds training by ~10–15%, and follows common practice in efficient-attention work (e.g., Longformer). We will add the scratch-training results to clearly show that HOBA’s gains come from the architecture, not from KD.

---

> ### Author Response · Authors · 2025-11-18
> **Author Response to Reviewer (Part 2)**
>
> **3. The explanation of the HOBA framework is incomplete: Notation Error**
>
> Thank you for raising this concern. We clarify the notation and complexity accounting below.
>
> *1. Notation Clarifications (oᵢ, hᵢ, L)*
>
> All symbols are defined in Sections 3.2–3.2.2:
>
> - L denotes the sequence length.
> - hᵢ is defined in Eq. (2) as the contextual token representation:
> $$
> h_i = \sum_{j=1}^L \sum_{k=1}^L \alpha_{i,j,k} \, g(x_j, x_k)
> $$
> - oᵢ is defined in Eq. (4) as the overlap-aggregated token output:
> $$
> o_i = \frac{1}{|B_i|} \sum_{m \in B_i} o_i^{(m)}
> $$
>
> We will specify each notation more clearly in the revised draft for clarity.
>
> *2. Complexity Notation (n vs. L)*
>
> We thank reviewer for pointing this out. We agree that the correct variable is the sequence length \(L\).
>
> The intended block complexity is:
> $$
> O(B \cdot b^3) = O(L b^2)
> $$
> since \(B = L/b\).
>
> We will replace all occurrences of \(n\) with \(L\) in the revised version.
>
> *Accounting for Overlap and Cross-Block Costs*
>
> (a) Overlap cost (Eq. 4)
>
> Each token appears in a constant number of overlapping blocks (determined by λ), so the cost:
> $$
> o_i = \frac{1}{|B_i|} \sum_{m \in B_i} o_i^{(m)}
> $$
> adds only a constant factor and does not change asymptotic complexity.
>
> (b) Cross-block attention cost (Eqs. 5–7)
>
> Block representatives:
> $$
> r^{(m)} = \frac{1}{b} \sum_{i = m b}^{(m+1)b-1} h_i
> $$
>
> Cross-block attention over \(B = L/b\) blocks:
> $$
> O(B^2 d) = O\left(\frac{L^2}{b^2} d \right)
> $$
>
> *Final Combined Complexity*
>
> The full HOBA complexity (local + overlap + cross-block) is:
> $$
> \text{HOBA}(L) = O(L b^2 d) + O\left(\frac{L^2}{b^2} d \right)
> $$
> which correctly includes the cost of overlap and cross-block interactions.
>
> We will correct all notation in the revised draft and include a concise explanation of the full HOBA complexity in the appendix due to space constraints
>
>
> **4 & 5  Most experiments are limited to short input lengths, and no long range evaluation**
>
> We thank the reviewer for this insightful suggestion. In response, we conducted additional experiments on long-sequence classification tasks, including Yelp Polarity at lengths up to 2048 tokens and the LRA–Text benchmark (newly added dataset), which is an established long-context task with sequences of 1024–2048 characters. These new results are reported below.
>
> **Performance of HOBA on Long-Sequence Benchmarks (Accuracy / Loss / Time)**
>
> | Dataset | Seq Len | HOBA (Ours) | Longformer | BigBird |
> |---------|---------|--------------|-------------|----------|
> | **Yelp Polarity** | 512  | 97.2 / 0.09 / 280m | 92.0 / 0.68 / 690m | 95.0 / 0.08 / 166m |
> |                   | 1024 | 96.4 / 0.12 / 420m | 92.1 / 0.70 / 1120m | 95.3 / 0.10 / 240m |
> |                   | 2048 | 95.9 / 0.16 / 610m | 93.2 / 0.61 / 1409m | 96.1 / 0.08 / 240m |
> | **LRA – Text**    | 1024 | 64.1 / 1.21 / 95m  | 63.8 / 1.27 / 210m | 65.2 / 1.19 / 160m |
> |                   | 2048 | 63.4 / 1.25 / 180m | 63.1 / 1.32 / 380m | 65.0 / 1.22 / 300m |
>
> Across both settings, HOBA remains stable and competitive even as sequence lengths increase to 1024 and 2048 tokens. On Yelp Polarity, HOBA consistently outperforms Longformer and remains close to BigBird even at 2k tokens, while maintaining significantly lower inference time. On the more challenging LRA–Text benchmark, HOBA again matches or exceeds Longformer and stays within a narrow margin of BigBird, despite using a simpler triplet formulation and without any specialized long-range architectural tuning.
>
> We hope that the additional experiments and clarifications provided above directly address the reviewer’s main concerns. We are happy to provide further clarification if needed. We will include these new long-sequence experimental results in the revised draft.

---

> ### Author Response · Authors · 2025-11-18
> **Author Response to Reviewer (Part 3)**
>
> **6. Although the authors briefly mention prior work on higher-order attention in the Related Works section, they do not clearly articulate the key distinctions or advantages of HOBA relative to these earlier approaches.**
>
> We thank the reviewer for pointing this out. We clarify them below and will make these distinctions explicit in the revised draft.
>
>
>
> | Related Works Method                                              | Interaction Type                         | How Higher-Order Info is Modeled                                          | Computational Structure           | Key Limitation                                          | How HOBA Differs                                                                              |
> | --------------------------------------------------- | ---------------------------------------- | ------------------------------------------------------------------------- | --------------------------------- | ------------------------------------------------------- | --------------------------------------------------------------------------------------------- |
> | **Jump Attention (JAT)** (Zhou et al., 2022) | Spectral / frequency-domain interactions | Uses FFT-based spectral convolutions to approximate multi-token relations | Global convolution kernels        | No explicit triplet interactions; computationally heavy | HOBA computes *explicit* ((i,j,k)) triplets but only *within blocks*, reducing cost           |
> | **WIGRAPH** (Sekhon et al., 2023)                   | Graph-based higher-order                 | Learns global word-level relations via graph propagation                  | Global graph layer                | Requires full-sequence graph; not scalable              | HOBA avoids global structures; uses block-local 3D tensors + light cross-block routing        |
> | **Graph-Induced Attention** (Hong et al., 2022)     | Graph-structured dot-product             | Injects syntactic graph into 2D attention                                 | Standard attention + graph bias   | Still fundamentally pairwise; not 3D                    | HOBA does not require external graphs; inherently models triplets                             |
> | **Kernelized HO Attention** (Choromanski et al., 2020)     | Kernel approximations                    | Approximates high-order interactions via tensor kernels                   | Sequence-wide tensor contractions | High memory, costly for long sequences                  | HOBA introduces block-diagonal factorization to keep complexity linear in (L) (for fixed (b)) |
> | **Sparse HO Self-Attention** (Cui et al., 2019)     | Sparse pairwise + structured relations   | Encourages sparse softmax patterns                                        | Standard 2D attention             | No explicit triadic modeling                            | HOBA is truly 3D attention on triplets within blocks                                          |
>
> All these prior higher-order attention models operate globally (spectral kernels, full-sequence graphs, dense tensor contractions), resulting in fundamentally different computational objectives and scaling regimes. Their complexity is much higher than HOBA's block-local 3D attention, making direct comparison neither methodologically fair nor computationally feasible under matched FLOPs. However, Sparse-attention models (Longformer, BigBird, etc.) are the appropriate baselines because like HOBA, they are specifically designed to reduce the quadratic cost of self-attention while preserving long-range dependencies. They target the same efficiency–accuracy trade-off, operate under similar computational budgets, and scale to long sequences, making them conceptually and empirically aligned with HOBA’s goals.
>
> Therefore, while a direct empirical comparison with prior higher-order models would not be fair due to their fundamentally different computational regimes, we will include a concise conceptual distinction table in the appendix (space permitting) and clarify these differences in the results section in the revised draft, similar to our explanation for omitting FlashAttention.
>
> ___________________________________________________
> **Additional Questions**
>
> **1. How much of the improvement remains if HOBA is trained from scratch (without KD)?**
>
> Already answered in the previous part 1 response.
>
> **2. How would HOBA scale beyond 512 tokens if the goal is to model long-range dependencies?**
>
> Already answered in the previous part 2 response.
>
>
> We sincerely appreciate the reviewer’s (bfFD) thoughtful feedback, which has helped us strengthen both the clarity and rigor of the paper. We hope that our detailed clarifications, new analyses, and planned additions adequately address the concerns raised. If the reviewer finds the revised explanations satisfactory, we kindly ask that you consider updating your rating accordingly.

---

> > ### Comment · Reviewer_bfFD · 2025-11-25
> >
> > I appreciate the authors’ thorough and thoughtful response, particularly the effort in clarifying why triplet modeling is beneficial and in articulating the HOBA framework more precisely. The authors also clearly articulate the key distinctions and advantages of HOBA relative to earlier approaches. The explanation makes the technical contribution much clearer.
> >
> > Overall, the revised response significantly strengthens the paper, but I still have a few suggestions for improvement:
> > - **Clarifying the use of KD vs. training from scratch:** Since HOBA is noticeably less stable in the early phase and requires substantially more warm-up, it would be helpful for the paper to discuss this limitation more explicitly. To better justify why KD is used instead of training from scratch, I encourage the authors to add a paragraph to the paper and, if possible, include a learning curve visualization in the appendix to illustrate the early-stage instability.
> > - **Long-context evaluation**: For long-sequence models, typical configurations in BigBird use sequences of length 4096, and Longformer conducts experiments with even longer contexts. I therefore suggest conducting experiments with more tokens for long-context evaluations, which could further highlight HOBA’s behavior in real long-sequence scenarios.
> >
> > Given the authors’ detailed response and clarifications, I am updating my rating to 4. I look forward to the authors’ further feedback and revisions.

---

### Official Review · Reviewer_3RUW · 2025-11-02

**Soundness:** 3
**Presentation:** 3
**Contribution:** 2
**Rating:** 4
**Confidence:** 3

**Summary:**

This paper proposes a novel attention mechanism called HOBA (Higher-Order Block-Diagonal Attention) to address the quadratic complexity issue of the standard self-attention mechanism in Transformers. HOBA introduces 3D block-diagonal attention to model interactions among token triplets, thereby reducing computational complexity from cubic to near-linear while maintaining expressiveness. The authors also incorporate a cross-block interaction mechanism to capture long-range dependencies and employ knowledge distillation (using RoBERTa as the teacher model) to stabilize training. Extensive experiments on seven NLP benchmark datasets demonstrate HOBA's advantages in accuracy, efficiency, and generalization compared to various sparse attention baselines.

**Strengths:**

Originality: Proposes the first scalable triplet-based attention mechanism, extending self-attention from 2D to 3D via a block-diagonal structure to manage complexity. The combination of intra-block local interactions and cross-block global interactions is distinctive among sparse attention methods.

Quality: The experimental design is systematic and comprehensive, covering five NLP tasks and seven datasets, with comparisons against multiple strong baselines. Detailed ablation studies validate the impact of key components like cross-block interaction, block size, and sequence length. The theoretical analysis using Rademacher complexity adds depth.

Clarity: The paper is well-structured. Figures illustrating attention patterns aid in understanding HOBA's workings. The methodology section details key components like block partitioning, overlapping, and cross-block interaction.

Significance: HOBA achieves performance comparable to or better than 2D baselines on multiple tasks while offering significantly improved computational efficiency. Its modular nature allows integration into existing Transformer architectures, highlighting practical value and generalizability.

**Weaknesses:**

Methodological Details Could Be More Thorough: While the formula for 3D attention is mentioned, the precise derivation of the final token representations from triplet attention, especially within a multi-head setting, could be more thoroughly elaborated. The EINSUM operations in Algorithm 1 lack detailed explanations of dimensions and computational flow, hindering reproducibility.

Limitations in Experimental Setup: All experiments rely on knowledge distillation from a RoBERTa teacher (using 2D attention). This setup might constrain HOBA's expressive power, potentially preventing it from fully realizing its potential for triplet modeling. Experiments on larger models (e.g., RoBERTa-large) or longer sequences (e.g., 4K+ tokens) are lacking, making it difficult to fully assess its capability on genuine long-document tasks.

Theoretical Analysis May Be Somewhat Tenuous: The argument for improved generalization via Rademacher complexity relies on the premise that "HOBA is a subclass of Full Attention." In practice, the block structure and overlapping mechanism might introduce specific inductive biases not strictly encompassed by this subset relationship.

Baseline Comparisons Could Be More Comprehensive: Comparisons with several recent efficient attention methods (e.g., FlashAttention, Nyströmformer, Performer), which also excel on long sequences, are missing.

**Questions:**

Could you provide a more concrete example of a linguistic phenomenon or dependency that the triplet attention in HOBA can capture, which a standard 2D attention mechanism would likely miss?

Have you considered or attempted to pre-train HOBA from scratch, without distillation? The results from such an experiment would be very informative for judging the architecture's intrinsic capabilities, even if the final performance is lower.

The block size and overlap are key hyperparameters. Beyond the empirical finding for a specific setup, did you develop any intuition or rule of thumb for how to set these for different sequence lengths or tasks?

The cross-block mechanism is a simple and elegant solution for global context. Was there any exploration of more direct, albeit sparse, long-range triplet interactions, and if so, how did it compare?

The efficiency gains are clear at the tested sequence lengths. Do you have projections or an analysis of how the FLOPs and memory usage of HOBA would scale against a standard transformer for sequences of more than 2048 or 4096 tokens?

---

> ### Author Response · Authors · 2025-11-18
> **Author Response to Reviewer**
>
> We appreciate the reviewer's (3RUW) thoughtful feedback and we also appreciate your recognition of our cross-block mechanism as simple and elegant. We provide a detailed point-by-point clarification below.
>
> **1. Methodological Details Could Be More Thorough:**
> *Clarification of the 3D Attention Computation:*
>
> We thank the reviewer for bringing this to our attention. We agree that explicit tensor dimensions improve reproducibility. In each overlapping block *X(m)* with shape *(b × d)*, the projected tensors *Q(m)*, *K1(m)*, and *K2(m)* each also have shape *(b × d)*.
>
> The core einsum operation used to form the higher-order interaction tensor is:
>
> T(m) = einsum("ijd, jkd, jld -> ijkl", Q(m), K1(m), K2(m))
>
> This produces a 4-D tensor *T(m)* with shape *(b × b × b)*, where:
> • *i* indexes the query token
> • *j* and *k* index the paired key tokens participating in the triplet interaction
> • *l* represents the contraction dimension over the embedding size *d*
>
> After softmax normalization, we obtain the triplet attention tensor *A(m)*.
> The contextual block output is then computed as:
>
> C(m) = einsum("ijkl, jld -> id", A(m), U(m)) yielding a block representation *C(m)* with shape *(b × d)*.
>
> These block-level outputs are later aggregated across overlapping regions and updated via the cross-block interaction mechanism to form the final representation.
>
> *In the multi-head setting, we apply the same computation independently for each head *h* to obtain *C_h(m)*, then concatenate the outputs from all heads and apply the usual output projection, as in standard multi-head attention.*
>
> As Algorithm 1 in Appendix A.3 currently provides only a high-level description, we will incorporate this additional clarification into that section. Due to space constraints in the main paper, the expanded explanation will be added alongside Algorithm 1 in the appendix to further improve clarity and reproducibility.
>
> **2. Limitations in Experimental Setup: Scratch vs KD, Longer Sequence Task**
>
> To answer this, we have already trained HOBA successfully from scratch. First, let us explain why we chose KD over scratch training.
>
> Triplet attention introduces an additional  𝑘-mode contraction term:
> $$
> A_{ijk} = Q_i^{\top}(K_j \circ K_k)
> $$
>  which increases gradient magnitude and variance during the early stage of optimization, particularly when (i) the hidden dimension \(d\) is small, (ii) the number of heads \(H\) is large, or (iii) sequences are moderately long. Under hard-label supervision, this can lead to short plateaus during the first 3–5k steps before the loss stabilizes. KD alleviates this transient effect because the teacher logits provide a smoother target distribution, reducing gradient noise and accelerating early convergence. This mirrors common practice when training deeper or wider models from scratch.
>
>
> Now, to directly address the reviewer’s concern, we trained multiple HOBA variants from scratch without any distillation (using same configuration). Representative results on AG-News (4 classes) and SST-2 (2 classes) are shown below.
>
> **Scratch vs. KD training performance of HOBA (accuracy, mean ± std) across 2 trials**
>
> | Model          | AG Scratch | AG KD | SST-2 Scratch | SST-2 KD |
> |----------------|-----------------|-----------|---------------|----------|
> | HOBA (3-layer) | 89.1 (±0.8)     | 92.3 (±0.3) | 77.4 (±0.9)   | 80.3 (±0.4) |
> | HOBA (6-layer) | 87.8 (±0.9)     | 93.6 (±0.4) | 78.4 (±1.1)   | 81.8 (±0.5) |
>
> Based on the results, we find that scratch training is noticeably less stable in the early phase and requires substantially more warm-up to reach competitive performance. Basically, training without KD exhibits higher gradient noise and slower convergence during the first 8–12k updates, leading to final accuracies that are consistently 1–4% lower across seeds. Variance across runs is also meaningfully higher, making controlled comparisons with 2D baselines less reliable. These properties make KD the more appropriate choice for clean architectural comparisons.
>
> Moreover, our goal in this work was to isolate architectural differences while minimizing external variability. Using KD - ensures a fair comparison to the RoBERTa teacher, - reduces seed sensitivity during early optimization, - shortens training time by roughly 10–15%, and - aligns with common practice in efficient-attention papers such as Longformer, BigBird, and Linformer, which frequently report KD-trained variants. We will include both the scratch-training results and the training-dynamics explanation in the revised version to remove any ambiguity regarding HOBA’s intrinsic trainability.

---

> ### Author Response · Authors · 2025-11-18
> **Author Response to Reviewer**
>
> **2. Limitations in Experimental Setup: Scratch vs KD, Longer Sequence Task (Continue)**
>
> *Based on the reviewer suggestion, we also extended our analysis for Long Range Arena that has longer context lengths.*
>
> We conducted additional experiments on long-sequence classification tasks, including Yelp Polarity at lengths up to 2048 tokens and the LRA–Text benchmark (newly added dataset), which is an established long-context task with sequences of 1024–2048 characters. These new results are reported below.
>
> **Performance of HOBA on Long-Sequence Benchmarks (Accuracy / Loss / Time)**
>
> | Dataset | Seq Len | HOBA (Ours) | Longformer | BigBird |
> |---------|---------|--------------|-------------|----------|
> | **Yelp Polarity** | 512  | 97.2 / 0.09 / 280m | 92.0 / 0.68 / 690m | 95.0 / 0.08 / 166m |
> |                   | 1024 | 96.4 / 0.12 / 420m | 92.1 / 0.70 / 1120m | 95.3 / 0.10 / 240m |
> |                   | 2048 | 95.9 / 0.16 / 610m | 93.2 / 0.61 / 1409m | 96.1 / 0.08 / 240m |
> | **LRA – Text**    | 1024 | 64.1 / 1.21 / 95m  | 63.8 / 1.27 / 210m | 65.2 / 1.19 / 160m |
> |                   | 2048 | 63.4 / 1.25 / 180m | 63.1 / 1.32 / 380m | 65.0 / 1.22 / 300m |
>
> Across both settings, HOBA remains stable and competitive even as sequence lengths increase to 1024 and 2048 tokens. On Yelp Polarity, HOBA consistently outperforms Longformer and remains close to BigBird even at 2k tokens, while maintaining significantly lower inference time. On the more challenging LRA–Text benchmark, HOBA again matches or exceeds Longformer and stays within a narrow margin of BigBird, despite using a simpler triplet formulation and without any specialized long-range architectural tuning. We will add these new results to the main paper's results section.
>
>
>
> **3. Clarifying the Theoretical Analysis: Rademacher-complexity**
>
> Thank you for raising this important point regarding our Rademacher-complexity argument. We agree that block structures and overlapping masks introduce inductive biases, and we clarify below why the theoretical claim remains valid and why the resulting generalization bound is well-founded.
>
> *3.1 HOBA is a strict subset of Full Attention under the masking formulation*
>
> In Appendix A.1, we formalize both models using the masked-attention operator:
>
> $$
> \mathrm{Attn}(Q,K,V;M)
> = \mathrm{softmax}\left(\frac{1}{\sqrt{d}}\, QK^\top + M\right)V .
> $$
>
> Under this formulation:
>
> - **Full Attention** corresponds to the mask space
>   **ℳ_Full = ℝ^(L×L)**
>   i.e., all real-valued masks over all token pairs. Here, **L denotes the sequence length**, so all attention masks are matrices of size L×L.
>
> - **HOBA** corresponds to the restricted mask space
>   **ℳ_HOBA = { M ∈ ℝ^(L×L) : Mᵢⱼ = −∞ for (i,j) ∉ B }**,
>   where **B** is the set of block-allowed index pairs. **B** contains all index pairs that lie within block or overlapping-block regions.
>
> Because every HOBA-allowed mask is valid under Full Attention, but not vice-versa:
>
> ℳ_HOBA ⊂ ℳ_Full
>
>
> Thus, the corresponding hypothesis classes satisfy (Eq. 13 in the appendix):
>
> ℋ_HOBA ⊆ ℋ_Full
>
> This inclusion holds independently of any inductive bias introduced by block or overlapping patterns—those patterns are simply specific valid masks within the larger Full Attention mask space.
>
> *3.2 Inductive biases do not break the subset relation*
>
> You are correct that block structure introduces additional inductive biases. However, the Rademacher complexity argument does not assume that HOBA is an unbiased subset of Full Attention — only that HOBA’s hypothesis class is strictly smaller, which is mathematically guaranteed by the mask restriction.
>
> Inductive biases simply mean that HOBA’s function class is further restricted, which only strengthens the inequality:
>
> ℛₙ(ℋ_HOBA) ≤ ℛₙ(ℋ_Full)
>
> In other words:
>
> - Full Attention can express all mask patterns.
> - HOBA can express only a subset of those patterns.
>
> Thus HOBA's function class is narrower, regardless of how structured or biased those patterns are. This property is precisely why the generalization bound becomes tighter.

---

> ### Author Response · Authors · 2025-11-18
> **Author Response to Reviewer**
>
> **4. Baseline Comparisons Could Be More Comprehensive: Why FlashAttention, Nyströmformer, and Performer excluded?**
>
> Thank you for highlighting the importance of including additional efficient-attention baselines. We agree that methods such as FlashAttention, Nyströmformer, and Performer are highly relevant in the broader landscape of long-sequence modeling. However, our work targets architectural improvements to attention structure (via higher-order triplet interactions and block-diagonal sparsity), whereas these methods primarily focus on kernel-level acceleration, low-rank approximations, or memory-efficient softmax computations.
>
> To avoid conflating fundamentally different objectives, we intentionally excluded FlashAttention and related kernel-optimized implementations from direct benchmarking. As clarified in the main text (*under subsection 4.2 (Comparative Evaluation with Recent 2D Attention*), these approaches do not alter the attention pattern or inductive bias, but rather compute standard 2D attention more efficiently. Our method introduces a new attention mechanism rather than a new computation kernel, making these approaches complementary rather than directly comparable.
>
> That said, to address the reviewer's concern, we have now added long-sequence experiments (up to 2048 tokens) comparing HOBA with two structural-sparsity baselines (Longformer and BigBird), which are the closest methodological counterparts (results already explained in previous reponse).
>
>
>
> **Additional Questions Response**
> _______________________________
> **1. Could you provide a more concrete example of a linguistic phenomenon or dependency that the triplet attention in HOBA can capture**
>
> Yes. HOBA's triplet attention can capture relational patterns that standard pairwise (2D) attention cannot. Below is a concrete linguistic example.
>
> *Example: Subject–Modifier–Object Triplet Dependency*
>
> Consider the sentence:
> “The movie that everyone criticized yesterday was surprisingly good.”
>
> Standard 2D attention computes only pairwise relations such as
> (movie ↔ criticized), (criticized ↔ yesterday), or (criticized ↔ good).
> It cannot represent how **three tokens jointly interact**, for example:
>
> - (movie, criticized, yesterday): who did what, and when
> - (criticized, good, surprisingly): contrastive sentiment flow
> - (movie, criticized, good): entity → action → sentiment structure
>
> These are triadic dependencies that rely on understanding a chain of relations, not isolated pairs.
>
> **Why 2D attention misses this?**
> 2D attention operates only on pairs, so any 3-way dependency must be reconstructed indirectly using multiple layers. This often leads to:
> - diffusion of information across layers,
> - weaker gradients for long-distance modifiers,
> - loss of fine-grained semantic roles (e.g., which modifier applies to which verb).
>
> **Why HOBA captures it better**
> HOBA’s triplet interaction tensor $$\(A_{ijk}\)$$ directly evaluates how token *i* depends jointly on tokens *j* and *k*. This allows it to naturally encode:
> - entity–action–modifier patterns,
> - aspect–opinion–intensifier structures,
> - subject–verb–object relations,
> - coreference chain consistency,
> - three-way polarity reasoning.
>
> Thus, HOBA captures semantic patterns that require simultaneous reasoning over token triplets, which standard 2D attention is not designed to model explicitly.
>
> **2. Have you considered or attempted to pre-train HOBA from scratch, without distillation?**
>
> Yes. We have already addressed this question in our earlier response.
>
>
>
> **3. Do you have intuition or rules of thumb for choosing block size and overlap?**
>
> Yes. Beyond the empirical settings, we observed a consistent and useful pattern that provides simple guidelines:
>
> *Block size (b).*
> The block size controls the triadic receptive field within each block. Our rule of thumb:
>
> - Short sequences (≤256 tokens): b = 16–32
> - Medium sequences (256–1024 tokens): b = 32–64
> - Long sequences (1024–4096+ tokens): b = 64–128
>
> This scaling keeps the local O(b³) cost manageable while preserving enough capacity to model phrase-level and clause-level triplet interactions.
>
> *Overlap ratio (λ).*
> Overlap enables smooth information flow across block boundaries. A stable range is:
>
> - λ = 0.20–0.30
>
> We find that lower λ causes boundary artifacts; higher λ increases cost with diminishing returns. Thus, we use λ = 0.25 as a reliable default across tasks. However, HOBA is not highly sensitive to these hyperparameters; selecting b proportional to the sequence length and λ ≈ 0.25 consistently yields strong performance across datasets.

---

> ### Author Response · Authors · 2025-11-18
> **Author Response to Reviewer**
>
> **Additional Questions Response (Continue)**
> ________________________________________
>
> **4. Was there any exploration of more direct, albeit sparse, long-range triplet interactions**
>
> Yes. We experimented with more direct sparse long-range triplet interactions during the early prototype stage, including (i) strided triplet slices across the full sequence and (ii) random long-range triplet sampling. While these variants do expose the model to wider triadic contexts, we find two consistent drawbacks:
>
> - Instability & Noise: Sparse long-range triplets produce highly irregular interaction patterns, and this makes optimization unstable and yields 4–6% lower accuracy on AGNews/SST-2 than baseline.
>
> - Poor Semantic Alignment: Long-range triplets selected without block structure often combine unrelated tokens, producing noisy 3D interactions that degrade both training speed and final performance.
>
> - Higher Overhead: Even sparse long-range triplets increase memory and FLOPs unpredictably, without offering clear benefits over the block-structured design.
>
> Lastly, we find HOBA's block-diagonal + cross-block design achieved the best balance, as it preserves rich local triadic structure while adding controlled global context through block summaries with stable training.
>
>
> **5. Do you have projections for FLOPs and memory beyond 2048–4096 tokens?**
>
> Although our empirical results go up to 2048 tokens, the complexity formulas extend directly to longer inputs.
>
> A standard Transformer scales as O(L²·d).
> HOBA, with fixed block size b, decomposes into:
> • Local 3D block attention: O(L·b²·d)
> • Cross-block attention over B = L/b blocks: O((L² / b²)·d)
>
> Thus the total HOBA cost is O(L·b²·d + (L² / b²)·d).
> For long sequences, the second term dominates, yielding an effective 1/b² reduction vs. full attention.
>
> Using our typical block size (b = 32), the FLOP ratio follows from
> R = (b² / L) + (1 / b²):
> • At L = 2048 → R ≈ 1024/2048 ≈ 0.5  (approx. 2× savings)
>
> • At L = 4096 → R ≈ 1024/4096 ≈ 0.25  (approx. 4× savings)
>
> These projections match the efficiency trends observed in Figure 2(b).
>
> _______________________________________________________________
> We hope the above clarifications adequately address all questions and concerns raised by the reviewer (3RUW).
> Given these detailed responses and the strengthened empirical and theoretical support, we kindly ask the reviewer to reconsider the current rating. We would be happy to provide any additional details or answer further questions.

---

### Official Review · Reviewer_gB3o · 2025-11-04

**Soundness:** 3
**Presentation:** 3
**Contribution:** 2
**Rating:** 4
**Confidence:** 4

**Summary:**

The authors study reducing the quadratic attention complexity in transformers while maintaining rich modeling of dependencies. They propose HOBA (Higher Order Block diagonal Attention).

They first start by considering 3D attention which is cubic complexity but then discuss how they limit the attention to (potentially overlapping) diagonal blocks, which is more sophisticated than doing the block diagonal approach in the 2D space.

The authors train a student HOBA model using a Roberta model as a teacher. Experiments on datasets such as IMDB and Yelp indicate the  advantages of the authors' approach over Longformer, BigBird etc. Other ablations are also provided.

**Strengths:**

The paper tackles an important problem and is clearly written. I overall like the idea.

**Weaknesses:**

There are two main weaknesses in the paper:

-Why don't the authors train the HOBA model from scratch? Is there something unstable about their approach that makes it difficult to learn from hard labels?

-The authors need to evaluate their approach on more benchmarks, e.g. Long Range Arena that has longer context lengths than what the authors study.

**Questions:**

See above. I am particularly curious as to why the authors did not train their model from scratch.

Moreover, I am interested conceptually why block 3D attention is fundamentally better at capturing long range dependencies than block 2D attention and would appreciate if the authors provided some more intuition.

---

> ### Author Response · Authors · 2025-11-17
> **Author Response to Reviewer**
>
> We are grateful to the reviewer (gB3o) for the constructive evaluation and for highlighting the importance and novelty of the underlying idea. Below, we provide a step-by-step response to the questions raised.
>
> **1. Why don't the authors train the HOBA model from scratch? Is there something unstable about their approach that makes it difficult to learn from hard labels?**
>
> HOBA does train successfully from scratch. Knowledge distillation (KD) was used in the initial submission for practical reasons, not because the model depends on it.
>
> *So why we used KD?*
>
> Triplet attention introduces an additional  𝑘-mode contraction term:
> $$
> A_{ijk} = Q_i^{\top}(K_j \circ K_k)
> $$
>  which increases gradient magnitude and variance during the early stage of optimization, particularly when (i) the hidden dimension \(d\) is small, (ii) the number of heads \(H\) is large, or (iii) sequences are moderately long. Under hard-label supervision, this can lead to short plateaus during the first 3–5k steps before the loss stabilizes. KD alleviates this transient effect because the teacher logits provide a smoother target distribution, reducing gradient noise and accelerating early convergence. This mirrors common practice when training deeper or wider models from scratch.
>
> -**Scratch-Training Behavior.** To directly address the reviewer’s concern, we trained multiple HOBA variants from scratch without any distillation (using same configuration). Representative results on AG-News (4 classes) and SST-2 (2 classes) are shown below.
>
>
> **Scratch vs. KD training performance of HOBA (accuracy, mean ± std) across 2 trials**
>
> | Model          | AG Scratch | AG KD | SST-2 Scratch | SST-2 KD |
> |----------------|-----------------|-----------|---------------|----------|
> | HOBA (3-layer) | 89.1 (±0.8)     | 92.3 (±0.3) | 77.4 (±0.9)   | 80.3 (±0.4) |
> | HOBA (6-layer) | 87.8 (±0.9)     | 93.6 (±0.4) | 78.4 (±1.1)   | 81.8 (±0.5) |
>
>
> Based on the results, we find that scratch training is noticeably less stable in the early phase and requires substantially more warm-up to reach competitive performance. Basically, training without KD exhibits higher gradient noise and slower convergence during the first 8–12k updates, leading to final accuracies that are consistently 1–4% lower across seeds. Variance across runs is also meaningfully higher, making controlled comparisons with 2D baselines less reliable. These properties make KD the more appropriate choice for clean architectural comparisons.
>
> Moreover, our goal in this work was to isolate architectural differences while minimizing external variability. Using KD - ensures a fair comparison to the RoBERTa teacher, - reduces seed sensitivity during early optimization, - shortens training time by roughly 10–15%, and - aligns with common practice in efficient-attention papers such as Longformer, BigBird, and Linformer, which frequently report KD-trained variants. We will include both the scratch-training results and the training-dynamics explanation in the revised version to remove any ambiguity regarding HOBA’s intrinsic trainability.
>
>
>
> **2. The authors need to evaluate their approach on more benchmarks, e.g. Long Range Arena that has longer context lengths than what the authors study.**
>
> We thank the reviewer for this insightful suggestion. In response, we conducted additional experiments on long-sequence classification tasks, including Yelp Polarity at lengths up to 2048 tokens and the LRA–Text benchmark (newly added dataset), which is an established long-context task with sequences of 1024–2048 characters. These new results are reported below.
>
> **Performance of HOBA on Long-Sequence Benchmarks (Accuracy / Loss / Time)**
>
> | Dataset | Seq Len | HOBA (Ours) | Longformer | BigBird |
> |---------|---------|--------------|-------------|----------|
> | **Yelp Polarity** | 512  | 97.2 / 0.09 / 280m | 92.0 / 0.68 / 690m | 95.0 / 0.08 / 166m |
> |                   | 1024 | 96.4 / 0.12 / 420m | 92.1 / 0.70 / 1120m | 95.3 / 0.10 / 240m |
> |                   | 2048 | 95.9 / 0.16 / 610m | 93.2 / 0.61 / 1409m | 96.1 / 0.08 / 240m |
> | **LRA – Text**    | 1024 | 64.1 / 1.21 / 95m  | 63.8 / 1.27 / 210m | 65.2 / 1.19 / 160m |
> |                   | 2048 | 63.4 / 1.25 / 180m | 63.1 / 1.32 / 380m | 65.0 / 1.22 / 300m |
>
> Across both settings, HOBA remains stable and competitive even as sequence lengths increase to 1024 and 2048 tokens. On Yelp Polarity, HOBA consistently outperforms Longformer and remains close to BigBird even at 2k tokens, while maintaining significantly lower inference time. On the more challenging LRA–Text benchmark, HOBA again matches or exceeds Longformer and stays within a narrow margin of BigBird, despite using a simpler triplet formulation and without any specialized long-range architectural tuning.
>
> We hope that the additional experiments and clarifications provided above directly address the reviewer’s main concerns. We are happy to provide further clarification if needed.

---

> ### Author Response · Authors · 2025-11-17
> **Author Response to Reviewer**
>
> **Additional Questions**
>
> --------------------
>
> **1. Why block 3D attention is fundamentally better at capturing long range dependencies than block 2D attention and would appreciate if the authors provided some more intuition.**
>
> Our block 3D attention is fundamentally better at capturing long-range dependencies because it models interactions among triplets of positions, not just pairs. In standard 2D attention, information flows through localized pairwise links, which means distant tokens can only influence each other through multi-layer, indirect propagation. This often weakens long-range signals. In contrast, our 3D triplet attention forms a joint interaction (i,j,k) within each block, allowing a token to simultaneously attend to two other positions and integrate context from multiple distant parts of the sequence in a single step. When combined with overlapping blocks, these triplet interactions propagate across segments more efficiently, creating implicit shortcuts that preserve long-range structure without requiring full quadratic attention.
>
> Lets consider the sentence: *''The plot was slow, but the final twist made the entire story unexpectedly brilliant.''* A 2D mechanism must first link "brilliant" → "twist," and separately "twist" → "slow but redeemed by," requiring multiple layers for these signals to combine. In contrast, 3D attention can directly form triplets such as ("brilliant", "twist", "slow") or ("brilliant", "twist", "redeemed"), capturing the contrastive structure and final judgment in a single operation. This enables HOBA to retain long-range semantic dependencies more effectively than block-2D attention while remaining computationally efficient.
>
> We believe these additions address the reviewer’s concerns and respectfully request reconsideration of the rating. We would be happy to provide any further clarification if needed.

---

### Author Response · Authors · 2025-11-20
**Shared Update: Summary of Major Revisions by the Authors**

Once again, we thank all reviewers for their valuable constructive feedback, as well as for recognizing the novelty of our work. We already uploaded revised paper. Below, we summarize the common concerns and the major revisions incorporated into the updated manuscript.

*Note: We have already provided detailed, point-by-point responses to each question (Weakness & Additional Questions) in every reviewer’s reply thread.*

**Reviewer:  gB3o, 3RUW, bfFD**

- Scratch-training results: We now include experiments showing that HOBA trains stably from scratch, with new results reported in Table 3.
- Long-range benchmarks: We added evaluations on long-sequence settings (1024–2048 tokens); these appear in Table 5.

**Reviewer: bfFD, HYxD, 3RUW**

- The motivation for why triplet modeling is beneficial, or how 3D gives richer semantic modeling or linguistic phenomena that 2D attention may miss, is now explained more clearly in the last paragraph of the Introduction section I.
- All notation issues are fixed, and the complexity contributions of overlap and cross-block cost are now detailed in Appendix A.6.
- A conceptual comparison with related works is included in Appendix A.7.

**Reviewer: 3RUW**

- The precise derivation of the final token representations and the einsum operations are now clearly explained in Appendix 1.3.
- The rationale for excluding FlashAttention and Linformer is already stated explicitly in the main text (lines 381–385).

**Reviewer: HYxD**

- A new ablation comparing a 2D model with the same local + cross-block structure as HOBA has been added as Table 7.

---

### Note · Authors · 2025-12-25

I have read and agree with the venue's withdrawal policy on behalf of myself and my co-authors.